# A flagellate-to-amoeboid switch in the closest living relatives of animals

**Thibaut Brunet[1,2]\*, Marvin Albert[3], William Roman[4], Maxwell C Coyle[1,2], Danielle C Spitzer[2], Nicole King[1,2]\***

[1]Howard Hughes Medical Institute, Chevy Chase, United States; [2]Department of Molecular and Cell Biology, University of California, Berkeley, Berkeley, United States; [3]Department of Molecular Life Sciences, University of Zürich, Zurich, Switzerland; [4]Department of Experimental and Health Sciences, Pompeu Fabra University (UPF), CIBERNED, Barcelona, Spain

**Abstract** Amoeboid cell types are fundamental to animal biology and broadly distributed across animal diversity, but their evolutionary origin is unclear. The closest living relatives of animals, the choanoflagellates, display a polarized cell architecture (with an apical flagellum encircled by microvilli) that resembles that of epithelial cells and suggests homology, but this architecture differs strikingly from the deformable phenotype of animal amoeboid cells, which instead evoke more distantly related eukaryotes, such as diverse amoebae. Here, we show that choanoflagellates subjected to confinement become amoeboid by retracting their flagella and activating myosin-based motility. This switch allows escape from confinement and is conserved across choanoflagellate diversity. The conservation of the amoeboid cell phenotype across animals and choanoflagellates, together with the conserved role of myosin, is consistent with homology of amoeboid motility in both lineages. We hypothesize that the differentiation between animal epithelial and crawling cells might have evolved from a stress-induced switch between flagellate and amoeboid forms in their single-celled ancestors.

**\*For correspondence:**
t.brunet@berkeley.edu (TB);
nking@berkeley.edu (NK)

**Competing interests:** The authors declare that no competing interests exist.

## Introduction

Amoeboid (or crawling) cell motility is central to several key aspects of animal biology, including development (*Kardash et al., 2010*; *Barton et al., 2016*), immunity (*Kopf et al., 2020*; *Reversat et al., 2020*), and wound healing (*Lamouille et al., 2014*). Nonetheless, the origin of animal amoeboid cells has remained mysterious (*Fritz-Laylin, 2020*), in part because the closest living relatives of animals, the choanoflagellates (*Ruiz-Trillo et al., 2008*; *King et al., 2008*), have been thought to exist solely in a flagellate form (except while encysted [*Leadbeater, 2015*]). The nature of the protozoan ancestor of animals was a matter of debate as early as the 19th century, when the relationship between animals and choanoflagellates was still unknown (reviewed in *Brunet and King, 2020*). Haeckel originally proposed in 1874 that animals descended from amoebae that evolved coloniality and later acquired cilia (*Haeckel, 1874*). In contrast, Metchnikoff proposed in 1886 that animals originated from colonies of flagellated cells resembling modern choanoflagellates (*Metchnikoff, 1886*), a view that was incorporated into later interpretations of Haeckel's Gastraea hypothesis (*Nielsen, 2008*). An intermediate view (*Willmer, 1971*) was inspired by the discovery of protozoans such as *Naegleria*, which alternate between a flagellate and an amoeboid form (*Schardinger, 1899*; *Fulton, 1977*). Under this scenario, the protozoan ancestor of animals may have already contained the genetic programs required for the evolution of differentiated crawling cells (such as sponge archeocytes, cnidarian amoebocytes, and vertebrate white blood cells) and flagellated cells (such as sperm cells and epithelial cells, which often retain an apical cilium/flagellum and/or microvilli [*Willmer, 1971*]). (Here we use the terms 'amoeboid motility' and 'crawling motility'

interchangeably; in animal cells, 'amoeboid motility' is sometimes used to refer specifically to bleb-mediated crawling [see discussion in *Fritz-Laylin et al., 2018*]).

Subsequent phylogenetic analyses revealed that protozoans known to alternate between flagellate and amoeboid forms (such as *Naegleria*) belonged to branches of the tree of life that were far removed from animals (*Fritz-Laylin et al., 2010*), making their relevance to animal origins uncertain. A single report of a colonial choanoflagellate containing both flagellated and amoeboid cells appeared in 1882 (*Kent, 1882*), but was never corroborated (*Leadbeater, 2015*), raising questions about its validity. Instead, the diagnostic and seemingly universal cell architecture of choanoflagellates is that of a seemingly rigid, ovoid cell bearing an apical collar complex – a single flagellum surrounded by a microvillar collar. The close evolutionary relationship between animals and choanoflagellates (*Ruiz-Trillo et al., 2008*), coupled with the similarity of morula stage animal embryos to spherical colonies of choanoflagellates (*Dayel et al., 2011*), lent apparent support to Metchnikoff's hypothesis (*Nielsen, 2008*) and led us and others to infer that the amoeboid cell types of animals had evolved from ancestral flagellate cells after the establishment of multicellularity (*Nielsen, 2008*; *Cavalier-Smith, 2017*; *King, 2004*; *Hashimshony et al., 2015*). However, modern choanoflagellates doubtless differ in some respects from their last common ancestor with animals (*Sogabe et al., 2019*) and some close outgroups to choanoflagellates and animals produce amoeboid cells (*Ruiz-Trillo et al., 2008*; *Sebé-Pedrós et al., 2017*; *Suga and Ruiz-Trillo, 2013*; *Sebé-Pedrós et al., 2013b*; *Fritz-Laylin, 2020*) or alternate flagellate and amoeboid forms (*Hehenberger et al., 2017*; *Tikhonenkov et al., 2020*; *Fritz-Laylin et al., 2017b*), raising the possibility that the cellular machinery for cell crawling and flagellar swimming both predate the divergence of the choanoflagellate and animal lineages (*Arendt et al., 2015*).

Consistent with a pre-metazoan origin of cell crawling, cell biological and biochemical studies have revealed similar cellular structures and conserved molecules involved in cellular crawling in animals and protists. These include cellular protrusions frequently involved in locomotion: (1) filopodia: slender, finger-like protrusions containing bundles of actin filaments, which are found in animals cells, choanoflagellates, and filastereans (*Sebé-Pedrós et al., 2013a*); (2) pseudopods: broad, flat protrusions containing branched F-actin networks reticulated by the Arp2/3 complex downstream of the actin regulators SCAR and WASP, which are observed in mammalian neutrophils and chytrid fungi (*Fritz-Laylin et al., 2017b*); and (3) blebs: round protrusions devoid of F-actin which form by delamination of the plasma membrane from the actomyosin cortex under the influence of actin/myosin II cortex contractility. Blebs have been observed in animal cells (such as mammalian macrophages and zebrafish primordial germ cells) and in free-living amoebae such as *Entamoeba histolytica* and *Dictyostelium discoideum* (reviewed in *Paluch and Raz, 2013* and *Charras and Paluch, 2008*). The cellular and biochemical similarities among cellular protrusions from animals and protists are consistent with a possible pre-metazoan origin of all three types of cellular protrusions (*Fritz-Laylin, 2020*).

However, the discontinuous phylogenetic distribution of cellular protrusions involved in crawling motility has also been interpreted as evidence for convergent evolution of amoeboid cells (*Cavalier-Smith, 2017*), and genomes do not resolve the controversy because proteins involved in crawling motility can also fulfill crawling-independent functions. For example, even though neither blebs nor locomotory pseudopods had previously been observed in choanoflagellates, choanoflagellate genomes encode predicted regulators of pseudopod formation (Arp2/3, SCAR, and WASP) and of blebbing (F-actin and myosin II). In choanoflagellates, Arp2/3, SCAR, and WASP have been proposed to mediate the formation of phagocytic cups, which are involved in feeding and might be structurally similar to pseudopods, but do not contribute to locomotion (*Fritz-Laylin et al., 2017b*). Similarly, the actin/myosin II complex not only is involved in blebbing but also underlies the formation of the cytokinetic ring that mediates sister-cell separation during cell division in multiple opisthokonts, including in yeasts and mammals (*Pollard and Wu, 2010*). Thus, conservation in choanoflagellates of proteins required for crawling cell motility in animals is also consistent with an alternative hypothesis, in which these proteins were restricted to locomotion-independent functions in the last common choanozoan ancestor and were subsequently co-opted during the independent evolution of locomotory protrusions in animal crawling cells and amoeboid protists, as suggested by *Cavalier-Smith, 2017* and others.

As the sister group of animals, choanoflagellates are a potentially informative taxon for distinguishing between these two alternative hypotheses for the ancestry of animal crawling cells.

Although choanoflagellates have not previously been reported to convert into crawling cells, the phenotypic repertoire of modern choanoflagellates is likely not completely known. Notably, fundamental aspects of choanoflagellate biology have only been discovered in the past few years, including collective contractility (*Brunet et al., 2019*), sexual reproduction (*Levin and King, 2013*), and bacterial regulation of multicellularity (*Alegado et al., 2012*) and of mating (*Woznica et al., 2017*). Moreover, rare and transient episodes of cell deformation have been reported to precede cell division in certain types of choanoflagellates (*Leadbeater, 2015*). Here, we report on the discovery of bleb-mediated amoeboid mobility in choanoflagellates, which expands the known phylogenetic distribution of this cellular behavior and strengthens the evidence for its pre-metazoan origin.

## Results

### Confinement induces an amoeboid phenotype in *Salpingoeca rosetta*

We report here on our recent and serendipitous discovery of environmentally relevant conditions under which the choanoflagellate *Salpingoeca rosetta* transdifferentiates from a flagellated state into an amoeboid state. While growing *S. rosetta* under conditions in which the cells were physically confined between the growth medium meniscus and the plate surface, we observed that some cells transitioned from a flagellated state to an amoeboid state (*Video 1*). Although amoeboid cells had not previously been reported in choanoflagellates, physical confinement regulates amoeboid cell differentiation and crawling motility in a wide range of eukaryotic cells, including zebrafish embryonic cells (*Ruprecht et al., 2015*), mammalian mesenchymal cells (*Liu et al., 2015*), some chytrid fungi (*Fritz-Laylin et al., 2017b*), dictyostelid amoebae (*Srivastava et al., 2020*), and euglenoid algae (*Noselli et al., 2019*). Moreover, cell confinement is likely of ecological relevance for choanoflagellates, which have been detected in diverse granular microenvironments (including soils [*Geisen et al., 2015*], marine sediments [*McKenzie et al., 1997*; *Nitsche et al., 2007*], sands [*Tikhonenkov and Mazei, 2006*], and silts [*Tikhonenkov and Mazei, 2006*]) whose pore sizes range from 1 mm to <1 µm and extend below the range of typical choanoflagellate cell diameters (~2–10 µm) (*Leadbeater, 2015*). In addiiton, choanoflagellates might also encounter transient cell confinement during attempts at phagocytosis by other microeukaryotic predators, such as ciliates or large amoebae (*Kumler et al., 2020*).

To test whether cell deformation can induce the amoeboid phenotype, we used a tunable system for dynamic cell confinement (*Le Berre et al., 2014*) and imaged live *S. rosetta* cells before, during, and after confinement (*Figure 1A*). Single cells of *S. rosetta* confined in a space of 4 µm or more maintained the canonical flagellate phenotype, consistent with the cell body not being deformed (*Figure 1B–D*). On the other hand, confinement below 3 µm elicited an active response from the cells, which started dynamically extending and retracting protrusions within a few seconds (*Figure 1B–D*; *Figure 1—video 1*). In 2 µm confinement or less, most cells retracted their flagellum within minutes, thus acquiring a fully amoeboid phenotype (*Figure 1B–J*; *Figure 1—figure supplement 1A*; *Figure 1—video 1*). While microvilli initially persisted in amoeboid cells, the microvilli underwent progressive scattering and, eventually, resorption over the following minutes (*Figure 1—video 1*). Releasing confinement fully reversed the phenotypic switch (*Figure 1K–P*). Newly unconfined cells retracted their dynamic protrusions, regained a round shape, and regrew a flagellum close to the position of the original, retracted one (*Figure 1—*

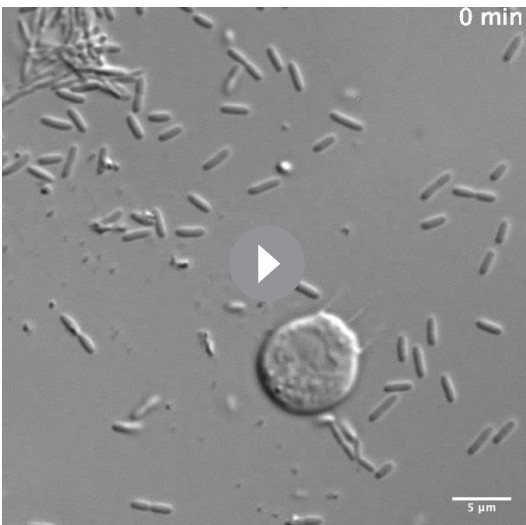

**Video 1.** Time-lapse of an *S. rosetta* cell undergoing progressive confinement by evaporation and switching to an amoeboid phenotype. The strain used was SrEpac and the starting cell type was slow swimmer. https://elifesciences.org/articles/61037#video1

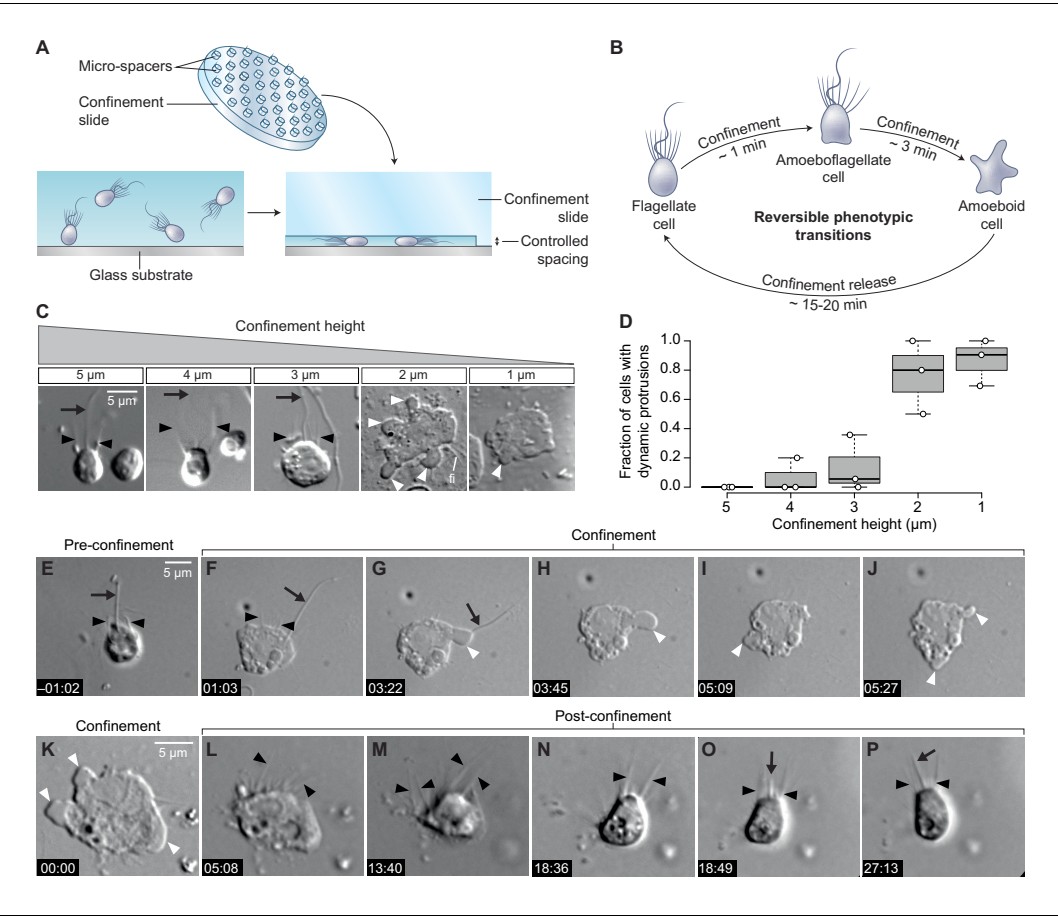

**Figure 1.** Confinement induces an amoeboid phenotype in the choanoflagellate *S. rosetta*. (**A**) Free-swimming cells (bottom left) were confined (bottom right) at a fixed height using confinement slides with micro-spacers (*Liu et al., 2015*; *Le Berre et al., 2014*) (top). (**B**) Confined *S. rosetta* cells underwent a rapid phenotypic transition, first from a flagellate form into an amoeboflagellate form, and eventually into an amoeboid form (that initially retains microvilli). Releasing confinement reversed this transition. (**C** and **D**) Confinement height correlated with the phenotypic switch. (**C**) Representative cells at each confinement height tested. (**D**) The flagellate form dominated at >3 µm confinement and the amoeboid form (defined by the presence of dynamic protrusions) at <3 µm. The number of cells (technical replicates) per batch (biological replicate) was as follows: 14, 6, and 12 cells for 5 µm confinement; 5, 5, and 11 cells for 4 µm confinement; 28, 18, and 6 cells for 3 µm confinement; 11, 5, and 6 cells for 2 µm confinement; and 13, 11, and 21 cells for 1 µm confinement. (**E–J**) Time series of an *S. rosetta* cell switching to the amoeboid form at 2 µm confinement. See *Figure 1—video 1* for multiple examples. (**K–P**) Time series of an amoeboid *S. rosetta* cell reverting to the flagellate form after release from confinement. See *Figure 1—video 2* for multiple examples. In all panels, white arrowheads indicate dynamic protrusions, black arrowheads indicate collar microvilli, and black arrows indicate the flagellum. Time stamps in black boxes shown as min:sec.

The online version of this article includes the following video and figure supplement(s) for figure 1:

**Figure supplement 1.** Flagellar retraction and regeneration during transitions between the flagellate and amoeboid forms.

**Figure supplement 2.** *S. rosetta* is competent to undergo the amoeboid switch in rosette and thecate forms.

**Figure 1—video 1.** Time-lapse of a population of *S. rosetta* cells before and during 2 µm confinement under a confinement slide controlled by a dynamic cell confiner.

https://elifesciences.org/articles/61037#fig1video1

**Figure 1—video 2.** Time-lapse of a population of *S. rosetta* cells before, during and after 2 µm confinement under a confinement slide controlled by a dynamic cell confiner.

https://elifesciences.org/articles/61037#fig1video2

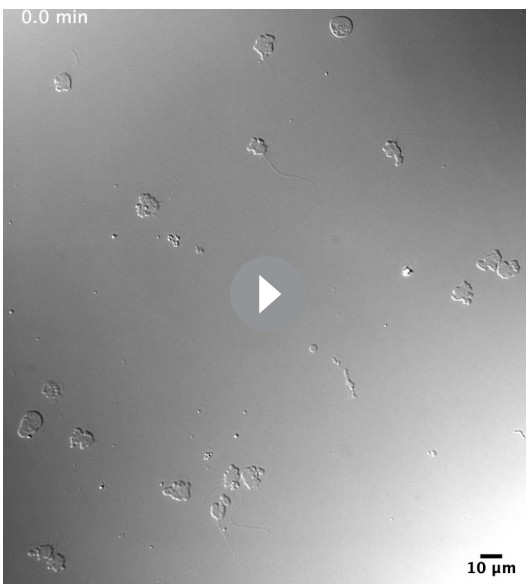

**Video 2.** Time-lapse of a population of *S. rosetta* cells confined between two glass cover slips using 2 μm microbeads as spacers. The strain used was SrEpac and the starting cell type was slow swimmer.
https://elifesciences.org/articles/61037#video2

*figure supplement 1B–D*; *Figure 1—video 2*), suggesting that information on apicobasal cell polarity is conserved in the amoeboid form, though not externally visible.

In addition to the solitary 'slow swimmer' cells used in the experiments above, *S. rosetta* can also differentiate into other cell types, including multicellular rosettes and sessile 'thecate' cells (*Dayel et al., 2011*). Both these cell types also differentiated into amoeboid cells with dynamic protrusions under confinement, showing that competence to undergo the amoeboid switch is not restricted to a single *S. rosetta* cell phenotype (*Figure 1—figure supplement 2*). Finally, *S. rosetta* responded in the same way to every type of confined environment tested, including the pressure-controlled dynamic cell confiner (*Figure 1A,C,D*), glass coverslips separated by microbeads, which served as spacers (*Video 2*), thin liquid films spread under a layer of oxygen-permeant oil (*Video 3*), and agar gels (*Video 4*). This suggests that the amoeboid switch is induced by cell deformation itself, independent of the properties of the substrate.

## *S. rosetta* amoeboid cells produce blebs

To reconstruct the evolutionary history of a given cellular phenotype (such as the amoeboid phenotype), an important pre-requisite is the identification of the cellular and molecular modules that underlie it in a phylogenetically relevant set of species (*Fritz-Laylin, 2020*; *Arendt, 2020*; *Carvalho-Santos et al., 2011*). Eukaryotic cell protrusions similar to those we observed in *S. rosetta* fall into

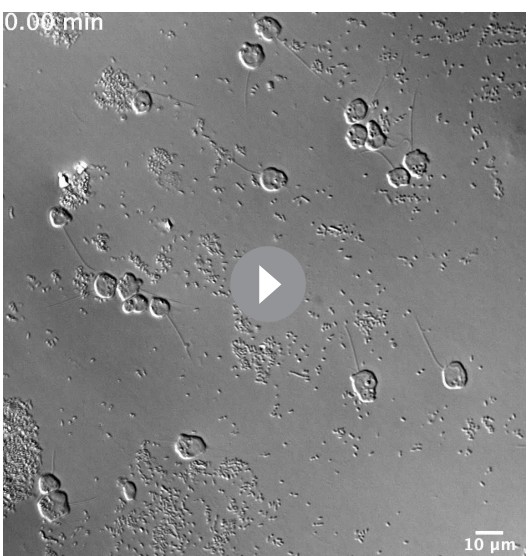

**Video 3.** Time-lapse of a population of *S. rosetta* cells confined in a thinly spread liquid film under a layer of anti-evaporation oil (see Materials and methods). The strain used was SrEpac and the starting cell type was slow swimmer.
https://elifesciences.org/articles/61037#video3

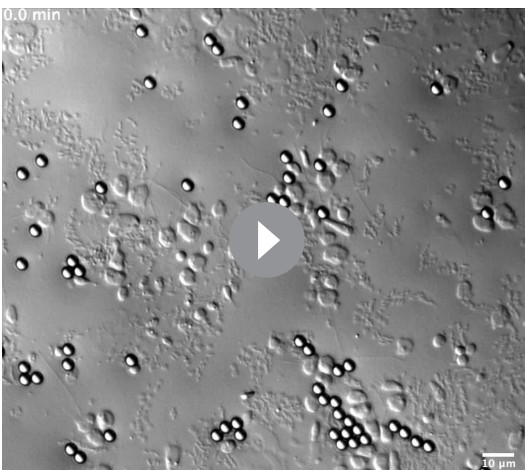

**Video 4.** Time-lapse of a population of *S. rosetta* cells confined together with microbeads on the surface of a 1% agar gel in artificial seawater, under a layer of anti-evaporation oil. The strain used was SrEpac and the starting cell type was slow swimmer.
https://elifesciences.org/articles/61037#video4

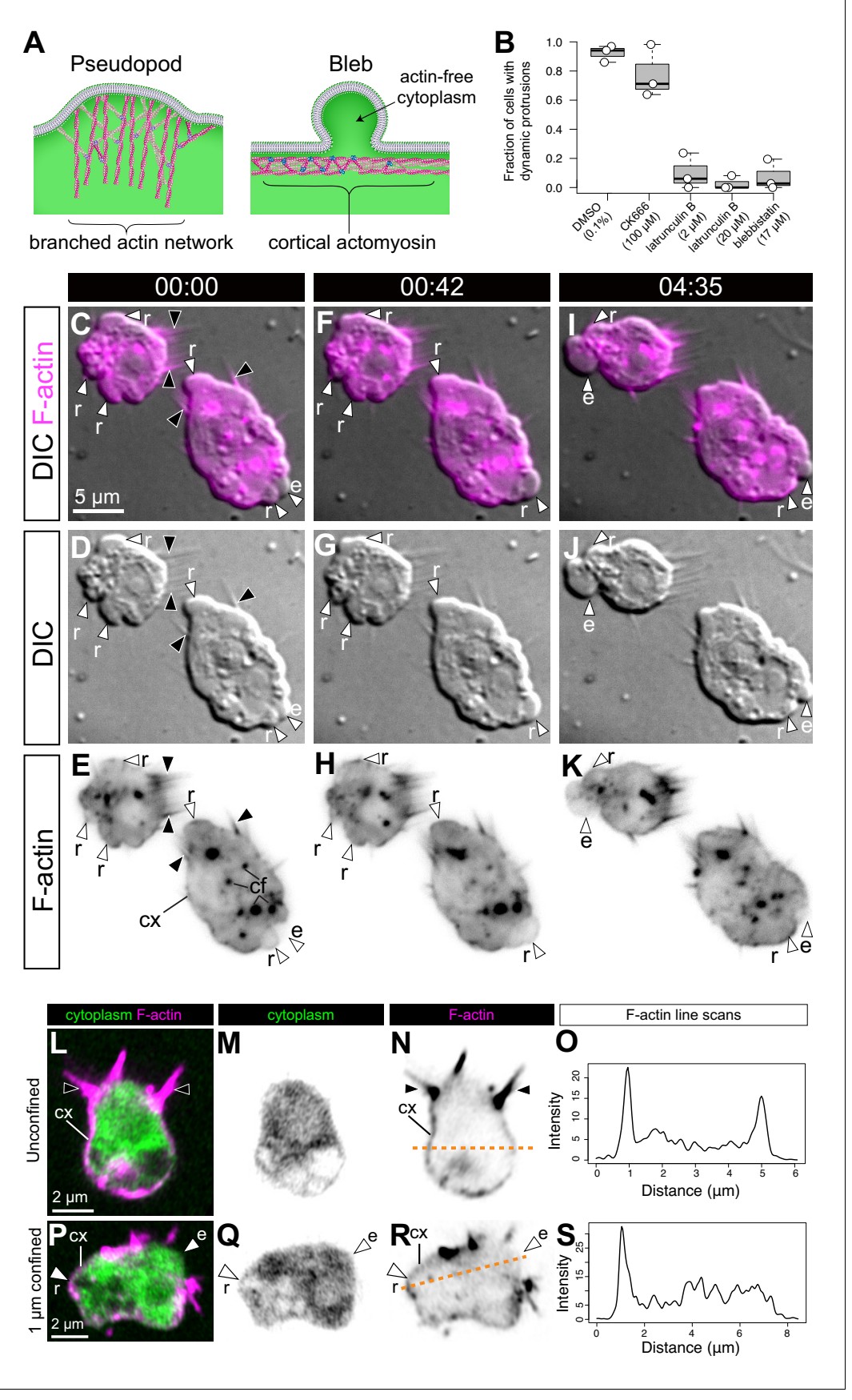

**Figure 2.** *S. rosetta* amoeboid cells generate blebs. (**A**) Protrusions in eukaryotic crawling cells can either be F-actin-filled pseudopods that form by polymerization of F-actin (pink) reticulated by the Arp2/3 complex (purple, left) or F-actin-free blebs that form through the action of contractile forces in the actomyosin cortex underlying the plasma membrane (right). The cytosol is green in both panels. Modified from *Fritz-Laylin et al., 2017a*. (**B**) Formation of dynamic protrusions required F-actin and myosin II activity, but not Arp2/3-mediated F-actin polymerization. Protrusions were abundant in DMSO-treated control cells (N = 32, 34, and 50 cells in the three respective biological replicates) and in cells treated with the Arp2/3 inhibitor CK666 (100 μM, N = 47, 56, and 59 cells in the three respective biological replicates) but virtually absent in cells treated with the F-actin polymerization inhibitor latrunculin B (at both 2 μM [N = 33, 76, and 11 cells in the three respective biological replicates] and 20 μM [N = 32, 49, and 22 cells in the three respective biological replicates]) or the myosin II inhibitor blebbistatin (17 μM, N = 36, 77, and 17 cells in the three respective biological replicates). This suggests that the dynamic protrusions were blebs. All cells were under 1 μm confinement. (**C–K**) Dynamic protrusions that form under confinement are blebs, as indicated by live imaging of two *S. rosetta* amoeboid cells expressing an F-actin marker (LifeAct-mCherry, magenta in C/F/I and black in E/H/K) (*Video 5*). (**C, F, I** and **E, H, K**) We observed that expanding blebs (e; defined as blebs that increased in size during the period of observation currently increasing in size) were cytoplasm-filled, but F-actin-free. We found that F-actin subsequently re-invaded blebs that then initiated retraction (r; defined as blebs that decreased in size during the period of observation). F-actin was also present as a cortical layer (cx), as in animal cells (*Chugh and Paluch, 2018*), and accumulated in cytoplasmic foci (cf). The cells were under 1 μm confinement. (**L–S**) *S. rosetta* cells fixed and stained for F-actin (phalloidin, magenta) and cytoplasm (FM 1–43 FX, which distributes to the cytoplasm of *S. rosetta* following fixation, green) confirmed that the protrusions of amoeboid cells initially lack F-actin and are therefore blebs. (L to N) A flagellate cell showing collar microvilli (black arrowheads) and F-actin cortex (cx). (**O**) Linescan of F-actin fluorescent intensity along the line of interest in (**N**), showing cortical actin as two peaks where the lines intersects the cells cortex. (**P–R**) An amoeboid cell showing both F-actin-free and F-actin-encased protrusions, respectively, interpreted as expanding (e) and retracting (r) blebs. (**S**) Linescan of F-actin fluorescent intensity along the line of interest in (**R**), showing cortical actin in the putative retracting bleb but not the putative expanding bleb. In all panels: white arrowheads: blebs, black arrowheads: microvilli, e: expanding blebs, r: retracting blebs, cf: cytoplasmic foci. Time stamps in black boxes shown as min:sec.

The online version of this article includes the following figure supplement(s) for figure 2:

**Figure supplement 1.** Latrunculin B and blebbistatin, but not CK666, inhibited the formation of dynamic cellular protrusions under cell confinement.

**Figure supplement 2.** Flow-through chamber for immunostaining of confined cells.

---

two different categories, pseudopods and blebs, that differ in their underlying mechanisms (see Introduction and *Fritz-Laylin et al., 2017a*). Pseudopods contain branched F-actin networks and have been best studied in adhesive animal mesenchymal cells (*Svitkina and Borisy, 1999*) but have also been identified in chytrid fungi (*Fritz-Laylin et al., 2017b*). By contrast, blebs are F-actin-free protrusions that form by delamination of the plasma membrane from the cortex under the influence of cortical actomyosin contractility (*Paluch and Raz, 2013*; *Charras and Paluch, 2008*; *Figure 2A*) and have been well documented in migratory primordial germ cells (*Kardash et al., 2010*). The presence of pseudopods and blebs is not mutually exclusive and multiple cells, including animal mesenchymal cells (*Bergert et al., 2012*), animal metastatic cells (*Sanz-Moreno and Marshall, 2010*), and dictyostelid amoebae (*Tyson et al., 2014*), can produce both.

To determine the nature of the *S. rosetta* cell protrusions, we treated cells with small molecule inhibitors of proteins required for pseudopod or bleb formation in animals and other amoeboid lineages. In animals, treatment of cells with latrunculin B inhibits the formation of F-actin networks and thereby prevents the formation of both pseudopods and blebs (*Li et al., 2011*; *Charras et al., 2005*). We found that inhibition of actin polymerization with latrunculin B prevented the formation of cell protrusions in *S. rosetta* (*Figure 2B*), thus implicating F-actin in their formation. Inhibition of the Arp2/3 complex with CK666 did not prevent formation of *S. rosetta* cell protrusions, suggesting they might represent blebs rather than pseudopods (*Figure 2B*). The inference that these protrusions are blebs was independently demonstrated by disruption of actomyosin contractility by treatment with the myosin II inhibitor blebbistatin, which reduced the formation of cell protrusions (*Figure 2B*; *Figure 2—figure supplement 1*).

To explore the level of similarity between choanoflagellate and animal cell blebs, we investigated two features that are diagnostic of animal cell blebbing: F-actin localization and dynamics and myosin II localization and contractility. In animal cells, nascent and expanding blebs are initially devoid of

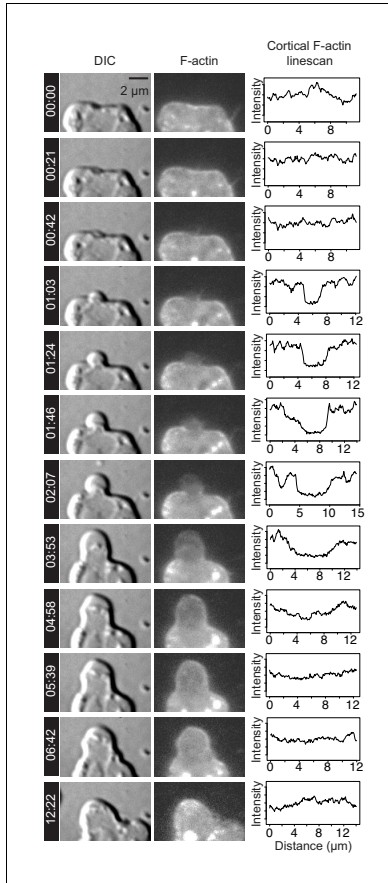

**Figure 3.** F-actin dynamics during the lifetime of a bleb. Time lapse imaging of a LifeAct-mCherry-expressing live cell (*Video 5*) by DIC (left column) and fluorescence microscopy (middle column) revealed membrane dynamics and actin localization during bleb formation. Line scans (right column) were used to quantify LifeAct-mCherry fluorescence (indicating relative F-actin levels, middle column) along the outline of the plasma membrane (visualized by DIC microscopy, left column). Bleb initiation (at 01:03) and expansion (from 01:03 to 03:53) correlated with relative reduction of F-actin within the expanding bleb. F-actin re-invaded the bleb to reassemble the cortex (04:58) prior to bleb retraction (06:42–12:22). Timestamps shown as mm:ss. Time points correspond to key events during blebbing and are not evenly spaced. X-axis indicates distance along linescan from left-to-right intersection of cell membrane with the bottom boundary of the image. Y-axis indicates relative intensity in arbitrary units (AU).

F-actin, which then re-invades the blebs before retraction (*Charras et al., 2006*). To investigate the localization and behavior of F-actin in cell protrusions, we generated a transgenic strain of *S. rosetta* expressing LifeAct-mCherry (*Booth et al., 2018*) and observed F-actin dynamics in the protrusions of confined cells that resembled those seen in blebbing animal cells (*Figure 2C–K*; *Figure 3*; *Videos 5–7*). Because LifeAct overexpression can sometimes create artifacts (*Courtemanche et al., 2016*), we also studied F-actin localization in fixed confined cells stained with fluorescent phalloidin (which stains F-actin) and a marker of the cytoplasm. Again, we observed both F-actin-free and F-actin-encased protrusions, consistent with our observations of expanding and retracting blebs, respectively (*Figure 2L–S*).

## Myosin II relocalizes to the cell cortex under confinement

Bleb formation in animal cells depends upon the contractile activity of myosin II (*Paluch and Raz, 2013*), which co-localizes with actin in the cell cortex and within retracting blebs (*Charras et al., 2006*). Because myosin II inhibition disrupted bleb formation (*Figure 2B*) in *S rosetta*, we hypothesized that cortical actomyosin might also underlie blebbing choanoflagellates. To investigate the intracellular distribution of myosin II in live flagellate and amoeboid *S. rosetta*, we generated an *S. rosetta* transgenic strain expressing a fluorescent myosin II fusion construct, Myosin Regulatory Light Chain-monomeric Teal Fluorescent Protein (MRLC-mTFP). In unconfined flagellate cells, MRLC-mTFP was diffusely distributed in the cytoplasm, while also forming a few cortical patches, most frequently at the basal pole of the cell (*Figure 4A*). Under 1 µm confinement, MRLC-mTFP redistributed in less than a minute into discrete foci and fibers, both cortical and cytoplasmic (*Figure 4B*). Quantification of fluorescence showed that this resulted in an increase in the cortical fraction of myosin II in confined cells (*Figure 4C*; *Figure 4D*, *Figure 4—figure supplement 1A*; *Figure 4—figure supplement 1B*; *Figure 4—figure supplement 1C*), similar to the confinement-induced redistribution of myosin to the cortex described in Dictyostelium (*Srivastava et al., 2020*) and vertebrate cells

(*Lomakin et al., 2020*; *Venturini et al., 2020*). In cells that were trapped at the border of the micropillars used for confinement and were thus only confined over part of their area, myosin II foci and fibers were only observed in the confined fraction of the cell (*Figure 4E*), suggesting that confinement and cellular deformation are sensed locally within cells (rather than by a central sensor such as the nucleus, as in vertebrate cells [*Lomakin et al., 2020*; *Venturini et al., 2020*]). Myosin II foci and fibers of confined cells underwent complex intracellular movements, possibly mediated by

contractility of the network (*Videos 8* and *9*). In blebbing cells, myosin II was absent from expanding blebs, but re-invaded blebs prior to retraction, similar to F-actin (*Figure 4F*; *Video 9* and *Figure 4— video 1*).

Most confined *S. rosetta* cells remained in one place and extended blebs in all directions without net locomotion. However, a few cells did migrate over short distances (about 15 µm; *Figure 5A-H*; *Video 1*, *Figure 1—video 1* and *Video 10*) with an initial median speed of 0.3 µm/min under 2 µm confinement, which decreased to 0.1 µm/min after about 10 min under confinement (*Figure 5E*; *Figure 1—video 1*). This decrease in speed correlated with a decrease in directional persistence (*Figure 5F*; *Figure 1—video 1*). Crawling with similar speed and persistence was also observed under 0.5 and 3.5 µm confinement (*Figure 5G–H*; *Figure 5—video 1*).

## Amoeboid cells retain cytoplasmic microtubules

Intracellular microtubule distribution regulates actomyosin activity in some animal cells (*Kopf et al., 2020*; *Chapa-y-Lazo et al., 2020*) and in Dictyostelium (*Sugiyama et al., 2015*), with microtubule-free zones experiencing higher local contractility and bleb retraction. In unconfined flagellated *S. rosetta* cells, we observed that cortical microtubules radiated from the apical basal body to form a cage underneath the entire plasma membrane, as previously reported (*Karpov and Leadbeater, 1998*; *Sebé-Pedrós et al., 2013a*; *Figure 4—figure supplement 2A–D*). In amoeboid cells, this cage remained present – but mostly detached from the plasma membrane and around the nucleus (*Figure 4—figure supplement 2E–H*). This is consistent with maintenance of the microtubule-organizing center and of apicobasal polarity in amoeboid cells (*Figure 1—figure supplement 1*). As interphase microtubules of choanoflagellates are resistant to standard inhibitors (e.g. nocodazole and colchicine), it was not possible to test directly whether microtubule dynamics regulates blebbing.

## Calcium is not required for the amoeboid switch in *S. rosetta*

Finally, we investigated whether confinement-induced bleb formation in choanoflagellates requires calcium signaling in choanoflagellates, as it does in animals (*Lomakin et al., 2020*; *Venturini et al., 2020*) and slime molds (*Srivastava et al., 2020*). To this end, we transferred the cells into calcium-free artificial seawater to deplete extracellular calcium and treated cells with BAPTA-AM to deplete intracellular calcium. Even after depletion of both intracellular and extracellular calcium, we observed no reduction in blebbing in response to confinement (*Figure 4—figure supplement 3*). Thus, despite the mechanistic similarities among choanoflagellates, animal cells, and slime molds in their reliance on actomyosin activity for bleb formation, calcium signaling does not appear to be required for confinement-induced blebbing in *S. rosetta*.

## The amoeboid switch is conserved across choanoflagellate diversity and allows escape from confined microenvironments

The amoeboid switch (at least as induced through confinement between glass slides) resulted in cells that extended blebs and seemed to probe their local environment, but only rarely migrated. However, we hypothesized that the crawling cell state might enable an escape response in a more complex environment, one resembling natural interstitial media. To test this hypothesis, we tracked the behavior of live cells in a heterogeneous environment containing zones of confinement surrounded by less confined spaces (*Figure 5I–J*; *Figure 5—video 1*). A matrix of PDMS pillars defined an environment in which choanoflagellates encountered 0.5 µm deep confinement zones surrounded by 3.5 µm deep spaces in which they could swim freely. Cells that were initially confined less than 15 µm away from a pillar border were capable of escaping to a 3.5 µm deep space (*Figure 5K*; *Figure 5— video 1*). Typically, a confined cell would first bleb irregularly and crawl slowly, until part of the cell – generally an expanding bleb – crossed the border of the pillar into the non-confined space. Following this, the cell would change shape and elongate away from the border of the pillar and crawl directionally until it had fully escaped (*Figure 5K–L*; *Figure 5—video 2*). Automated cell segmentation followed by morphometric quantification confirmed that escaping cells reliably elongated (*Figure 5M*; *Figure 5—figure supplement 1*; *Figure 5—video 2*), while non-escaping cells (that never detected a border) remained nearly round (*Figure 5N*), indicating that this escape behavior involves specific cell shape changes. Interestingly, the first blebs that crossed the border were

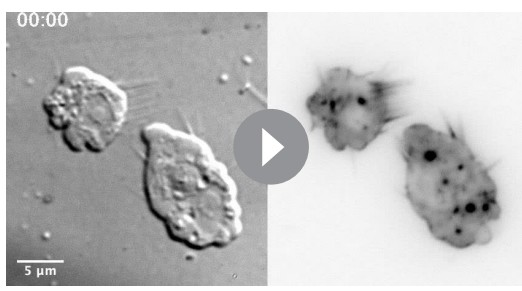

**Video 5.** Time-lapse of two *S. rosetta* cells under 1 µm confinement expressing LifeAct-mCherry (which marks F-actin). Blebs first form as cytoplasm-filled, F-actin-free protrusions and are re-invaded by F-actin before retraction.

https://elifesciences.org/articles/61037#video5

occasionally shed from the cell and reabsorbed during escape (*Video 11*). Finally, treatment with blebbistatin dramatically reduced the escape behavior (*Figure 5O*), consistent with this behavior requiring myosin activity. This type of behavior might allow choanoflagellates to escape from tightly packed silts (<3 µm granularity) into the water column or more loosely packed interstitial environments. It might also allow escape during attempts at phagocytosis by predators.

Having uncovered an amoeboid switch in *S. rosetta*, we sought to assess the phylogenetic distribution of the amoeboid switch across choanoflagellate diversity in order to determine whether this phenotype may have been present in the last common ancestor of choanoflagellates. We tested the effect of 2 µm confinement on six additional choanoflagellate species that together represent the main branches of the choanoflagellate phylogenetic tree (*Carr et al., 2017*). All displayed blebbing activity under confinement (*Figure 6A–J*) with the exception of *Diaphanoeca grandis* (*Figure 6K and L*; *Figure 6—video 1*) – which could indicate a secondary loss of the ameboid switch in this lineage – or that different conditions are required to induce the phenotype (such as <2 µm confinement). The amoeboid switch in the other five species covers a spectrum of phenotypes. The least pronounced response was seen in *Choanoeca flexa* (*Figure 6G–H*), whose sheet colonies (*Brunet et al., 2019*) dissociate into single cells that bleb and migrate over short distances without retracting their flagella, thus keeping an amoeboflagellate phenotype. Two other species, *Monosiga brevicollis* (*Figure 6E–F*; *Video 12*) and *Acanthoeca spectabilis* (*Figure 6I–J*; *Video 13*), showed a similar response to *S. rosetta*, including blebs and flagellar retraction. *Salpingoeca helianthica* generated very large and branched blebs (*Figure 6A,B*; *Video 14*), often longer than the rest of the cell body – reminiscent of the 'lobopods' described in some protists (*Tikhonenkov et al., 2020*). Finally, *Salpingoeca urceolata* (*Figure 6C–D*; *Video 15*) differentiated into amoebae capable of sustained migration over long distances (>40 µm).

## Discussion

The discovery of an amoeboid switch in a choanoflagellate has the potential to inform the ancestry of amoeboid cells in animals. Crawling cell motility is found in virtually all animal lineages and is important for embryonic development (e.g. in neural crest migration [*Berndt et al., 2008*], gastrulation [*Kraus et al., 2019*], and primordial germ cell migration [*Grimaldi and Raz, 2020*; *Barton et al., 2016*; *Kardash et al., 2010*]), wound healing (*Lamouille et al., 2014*), and immunity (e.g. in phagocytes patrolling tissues

**Video 6.** Time-lapse of an *S. rosetta* cell under 1 µm confinement expressing LifeAct-mCherry (which marks F-actin). Blebs first form as cytoplasm-filled, F-actin-free protrusions and are re-invaded by F-actin before retraction.

https://elifesciences.org/articles/61037#video6

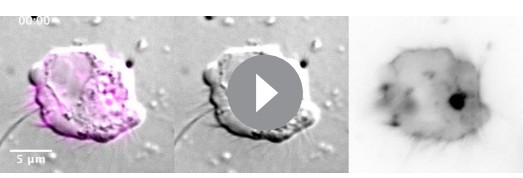

**Video 7.** Time-lapse of an *S. rosetta* cell under 1 µm confinement expressing LifeAct-mCherry (which marks F-actin). Blebs first form as cytoplasm-filled, F-actin-free protrusions and are re-invaded by F-actin before retraction.

https://elifesciences.org/articles/61037#video7

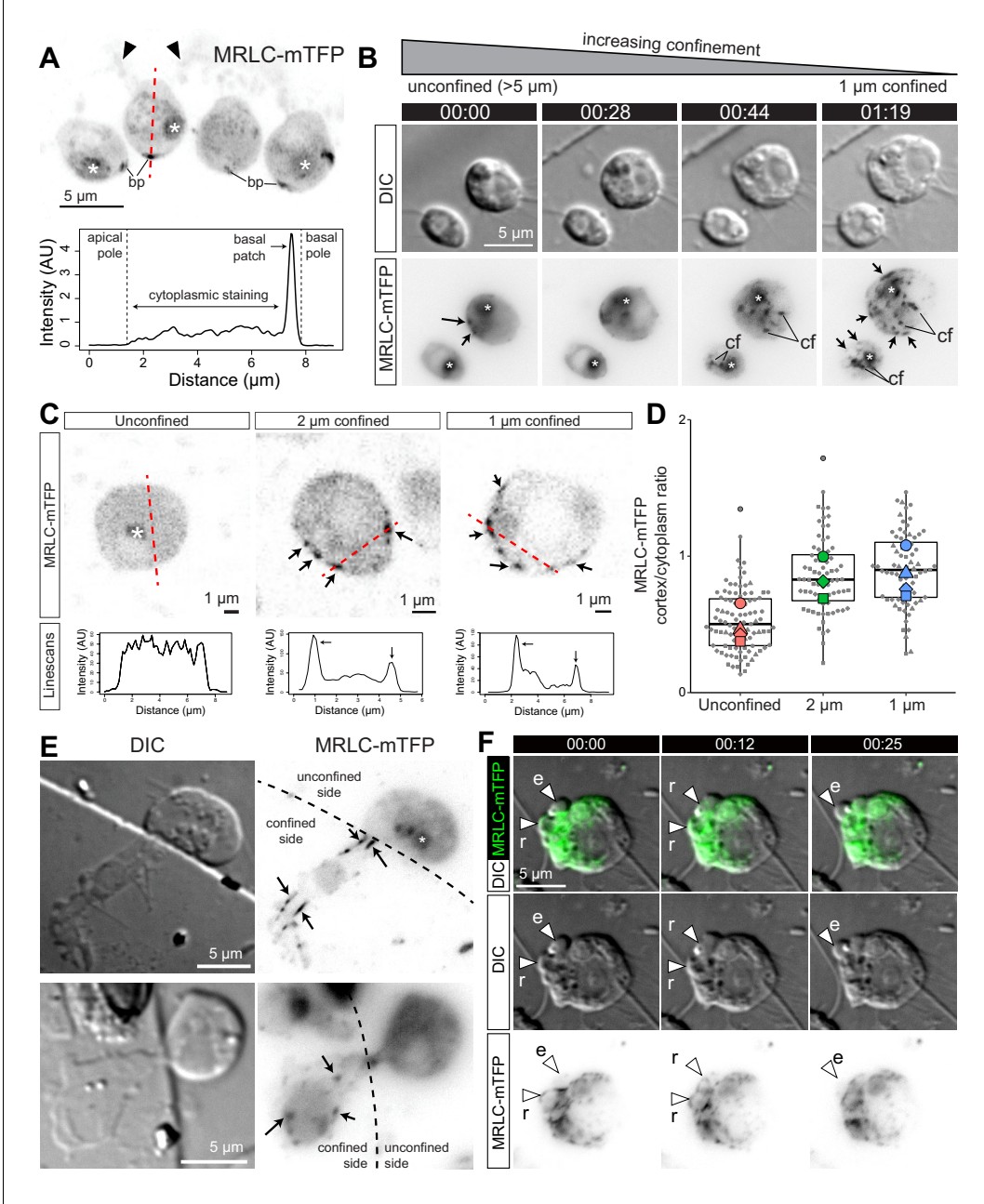

**Figure 4.** Myosin II undergoes intracellular redistribution in response to confinement. (**A**) Top: MRLC-mTFP fluorescence in four unconfined flagellate *S. rosetta* cells from a chain colony (*Dayel et al., 2011*). The position of the microvilli (black arrowheads) can be inferred from weak autofluorescence of bacteria captured by the collar. Asterisks (*) indicate fluorescent signal from the food vacuole (possibly due to autofluorescence of phagocytosed bacteria and/or to fluorescent protein degradation by autophagy; *Wetzel et al., 2018*). Most MRLC-mTFP is cytoplasmic but a cortical basal patch (bp) can be observed. Bottom: linescan of MRLC-mTFP fluorescence intensity along the line of interest (**A**; red dotted line), showing both diffuse cytoplasmic staining and cortical staining at the basal patch. (**B**) MRLC-mTFP is redistributed in less than a minute in response to confinement, from a diffuse cytoplasm staining into discrete cortical and cytoplasmic foci and fibers. Time-lapse of two MRLC-mTFP-expressing *S. rosetta* cells before, during and after establishment of 1 μm confinement. (cf): cytoplasmic foci, arrows: cortical foci and fibers, (*): food vacuole signal. (**C**) Top: confocal images of MRLC-mTFP in representative unconfined, 2 μm-confined and 1 μm-confined *S. rosetta*. Black arrows: cortical foci. See *Figure 4—figure supplement 1* for more cells. Bottom: line scans along the lines of interest in the top panels. Cortical foci of myosin II appear as peaks where the line of interest (red dotted line) intersects the cell cortex. (**D**) Myosin II is

*Figure 4 continued on next page*

*Figure 4 continued*

enriched at the cortex of confined cells, as manifested by an increase in the cortical/cytoplasmic ratio in MRLC-mTFP fluorescence. Results are depicted as a SuperPlot (*Lord et al., 2020*) where large dots are biological replicates (batches of cells) and small dots technical replicates (individual cells). p=$2.5 \times 10^{-2}$ and $1.3 \times 10^{-2}$ by Dunnett's test for comparison of multiple treatments to a control, for 2 and 1 μm confinement, respectively. (E) Half-confined cells (partly trapped under a micropillar used for confinement, see *Figure 1A*) only show condensation of myosin II into foci and fibers (arrows) in the confined part of the cell. (F) Localization of MRLC-mTFP in a blebbing amoeboflagellate cell under 1 μm confinement (*Video 9*; see also *Figure 4—video 1*). Myosin II is absent from expanding blebs (e; defined as in *Figure 2*) but re-invades retracting blebs (r).

The online version of this article includes the following video and figure supplement(s) for figure 4:

**Figure supplement 1.** Confinement results in redistribution of MRLC-mTFP from the cytoplasm to the cortex.
**Figure supplement 2.** Microtubules are present in *S. rosetta* amoeboid cells.
**Figure supplement 3.** The amoeboid switch is independent of calcium signaling.
**Figure 4—video 1.** Time-lapse of *S. rosetta* cells under 1 μm confinement expressing MRLC-mTFP (which marks myosin II) and imaged by epifluorescence microscopy.
https://elifesciences.org/articles/61037#fig4video1

---

[*Kopf et al., 2020*]). Cell crawling has also been frequently observed in the sister lineage of choanozoans, filastereans (*Tikhonenkov et al., 2020*; *Hehenberger et al., 2017*; *Sebé-Pedrós et al., 2013b*) – but its seeming absence from choanoflagellates had made its evolutionary history unclear. Together with the existing comparative evidence, our data suggest that the last common ancestor of choanoflagellates and animals had the ability to differentiate into amoeboid and amoeboflagellate cells under confinement.

Consistent with an ancient origin of all three phenotypes, amoeboid, flagellate, and amoeboflagellate cells are all broadly distributed in opisthokonts (animals, fungi, and their relatives; *Figure 6M*). Interestingly, amoeboflagellate phenotypes have recently been described in several species occupying key phylogenetic positions, including in the sister group of choanozoans (the filastereans [*Tikhonenkov et al., 2020*; *Hehenberger et al., 2017*]), in early-branching fungi (*Karpov et al., 2019*; *Karpov et al., 2018*; *Fritz-Laylin et al., 2017b*; *Galindo et al., 2020*), in the two closest known relatives of opisthokonts (apusomonads [*Cavalier-Smith and Chao, 2010*] and breviates [*Minge et al., 2009*]), and in early-branching amoebozoans (*Ptáčková et al., 2013*). This phylogenetic distribution is consistent with an ancient origin and broad conservation of the amoeboflagellate phenotype in opisthokonts (*Cavalier-Smith and Chao, 2003*).

One important question concerns the homology of cellular blebs between choanoflagellates and animals. Outside choanozoans, blebs have been well described in amoebozoans, notably *D. discoideum* (*Yoshida and Soldati, 2006*; *Merkel et al., 2000*) and *E. histolytica* (*Maugis et al., 2010*). Cell biological studies have revealed close structural and mechanistic similarities between amoebozoan and metazoan blebs, consistent with, but not demonstrative of, a pre-metazoan origin of blebbing (*Paluch and Raz, 2013*; *Charras and Paluch, 2008*). However, the absence of described blebs in intervening branches (fungi, ichthyosporeans, filastereans, and choanoflagellates) raised the alternative possibility that blebs might have evolved convergently in amoebozoans and animals by independent co-option of cortical actomyosin contractility, which could have been ancestrally involved in other functions (such as cytokinesis; see Introduction). By documenting blebs similar to those of amoebozoans and of animal cells in choanoflagellates, our study significantly extends the known phylogenetic distribution of blebbing.

Importantly, the likelihood of homology between two structures is not only a function of their similarity, but also depends on their phylogenetic distribution (*de and Mario, 1991*; *Remane, 1983*; *Wagner, 1989*). Because choanoflagellates and metazoans are sister groups, morphologically similar features that are also based on the same molecular machinery in these two groups are most parsimoniously interpreted as homologous, and our results thus provide evidence for a pre-metazoan origin of blebbing and of crawling motility. However, evolutionary convergence remains difficult to entirely rule out, given that the molecules known to be required for blebbing (such as actin and myosin II) have other functions. Future research in choanoflagellates, amoebozoans, and animals might help determine whether specific regulators of blebbing exist (similar to the ancient eukaryotic 'flagellar toolkit' [*Carvalho-Santos et al., 2011*]). Further comparative evidence might also come from

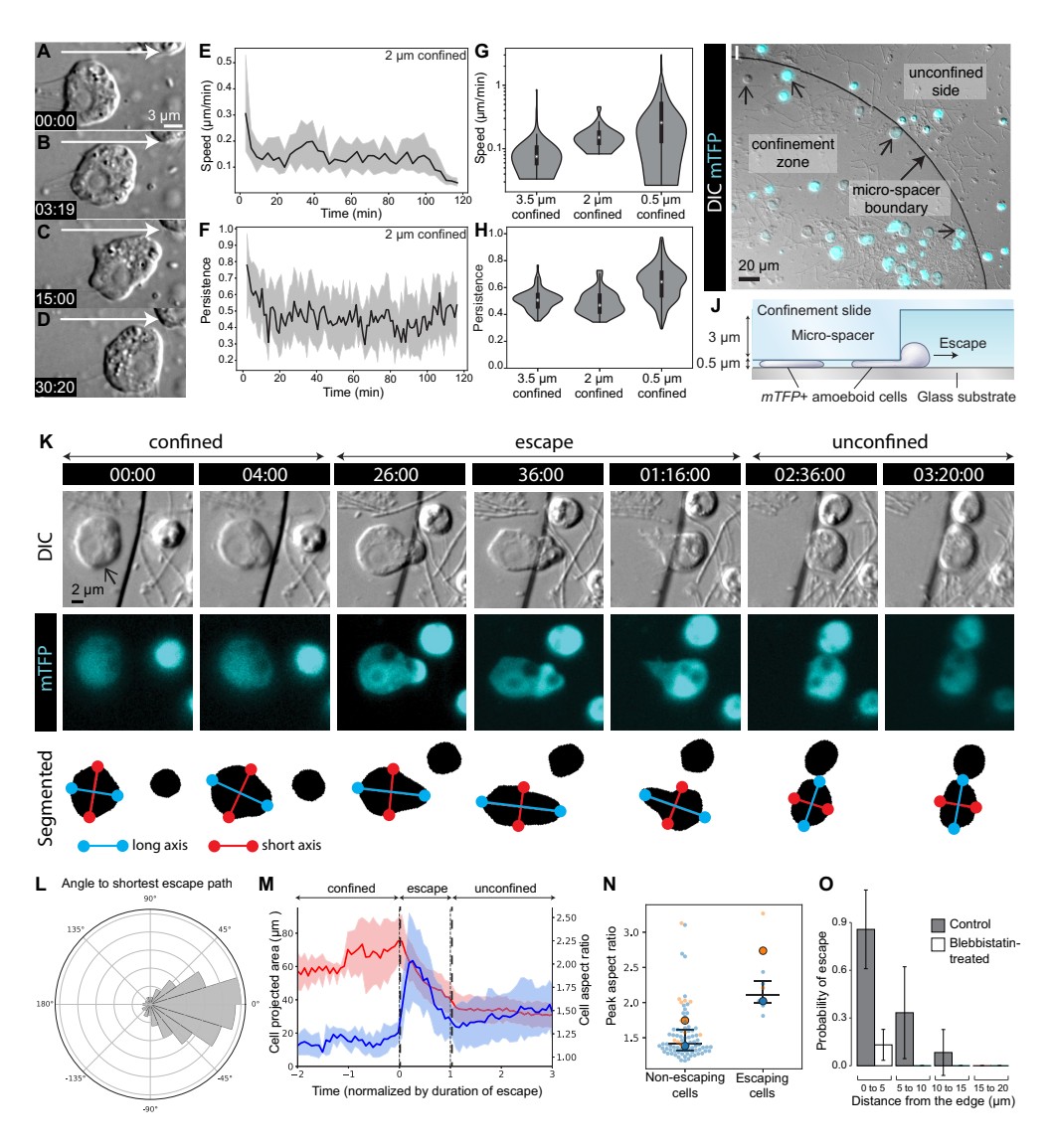

**Figure 5.** The amoeboid switch allows escape from confinement. (**A–D**) Amoeboid cell crawling after flagellar retraction (***Figure 1—video 1***). White arrow indicates direction of movement. (**E** and **F**) Speed and directional persistence of the cells in ***Figure 1—video 1*** (2 μm confinement). Directional persistence was defined as the ratio of the total path to the Euclidean distance over 2 min. (**G** and **H**) Violin plots showing speed and directional persistence measured over 100 min under 0.5 μm (***Figure 5—video 1***, non-escaping cells under the micropillar – see following panels), 2 μm (***Figure 1—video 1***), and 3.5 μm (***Figure 5—video 1***, cells outside the micropillar – see following panels) confinement. (**I**) mTFP-expressing *S. rosetta* cells (cyan; confined cells that escaped during the assay indicated with small arrows) distributed within and outside the confinement zone (border indicated with larger arrow) at the beginning of an escape assay (***Figure 5—video 1***). (**J**) Schematic of cross-section through escape assay set-up from (**I**). (**K**) Time series of an mTFP-expressing cell (arrow) during escape from confinement (top, DIC; middle, mTFP; bottom, segmentation of mTFP fluorescence to reveal cell shape; ***Figure 5—video 2***). Automated detection of the long (blue) and short (red) axes of the cell revealed that the cell elongated during crossing of the confinement border and relaxed into a more rounded shape once escape was complete. (**L**) Cells crawled directionally during escape. Bullseye diagram showing the distribution of angular differences between crawling and the shortest possible escape path in escaping cells (N = 8). (**M**) Escaping cells (N = 8) consistently elongated during escape and resumed a rounder shape once in the unconfined area. Escape also corresponded to a decrease in the projected area of the cell. Mean aspect ratio (red line) and projected area (blue line), ribbons: standard deviation. (**N**) Escaping cells acquired a highly elongated shape. Non-escaping cells did not reach comparable elongation levels, as indicated by the peak aspect ratio. Results are depicted as a SuperPlot

*Figure 5 continued on next page*

*Figure 5 continued*

(*Lord et al., 2020*) with biological replicates (time-lapse movies of a cell population) represented as large dots and technical replicates (individual cells within each movie) as small dots. Replicate 1 (blue dots) included 81 cells of which six escaped (p=2.2 × 10$^{-4}$ by Mann–Whitney's U test). Replicate 2 (orange dots) included 13 cells of which two escaped (p=3.0 × 10$^{-2}$ by Mann–Whitney's U test). (**O**) Escape required myosin II activity. Control cells (three biological replicates with N = 95, 35, and 12 cells) almost always escaped confinement if they were initially located less than 5 µm away from the border, and some escaped from as far as 15 µm. Seventeen micromolar blebbistatin-treated cells (two biological replicates with N = 110 and 73 cells) virtually never escaped. Time stamps in black boxes shown as min:sec.

The online version of this article includes the following video and figure supplement(s) for figure 5:

**Figure supplement 1.** Amoeboid cells elongate during escape from confinement and revert to a rounded shape after escape.

**Figure 5—video 1.** Time-lapse of an mTFP-expressing population of *S. rosetta* cells trapped in a 0.5 µm space under a circular micropillar.

https://elifesciences.org/articles/61037#fig5video1

**Figure 5—video 2.** Close-up of an escaping cell (from *Figure 5—video 1*) showing DIC channel (top left), mTFP channel (top right), segmented cell shape (bottom left), and segmented cell shape (magenta) overlaid with the DIC channel (gray).

https://elifesciences.org/articles/61037#fig5video2

---

investigation of the possibility of blebbing in other single-celled opisthokonts such as fungi, ichthyosporeans, and filastereans, for example (but not necessarily only) in response to confinement.

If blebs are indeed homologous in animals and choanoflagellates, opisthokont ancestors might have crawled using a combination of (1) blebs (as in amoeboid choanoflagellates, amoeboid animal cells, and amoebozoans [*Liu et al., 2015*; *Ruprecht et al., 2015*; *Srivastava et al., 2020*]), (2) pseudopods (which are present in choanoflagellates during phagocytosis and during crawling in chytrid fungi, amoebozoans, and some holozoans [*Srivastava et al., 2020*; *Tikhonenkov et al., 2020*; *Fritz-Laylin, 2020*; *Fritz-Laylin et al., 2017b*; *Leadbeater, 2015*]), and (3) filopodia (which contribute to locomotion in filastereans and during choanoflagellate settlement [*Sebé-Pedrós et al., 2013a*; *Dayel et al., 2011*]). Moreover, cell crawling is regulated by confinement in animal cells (*Liu et al., 2015*; *Ruprecht et al., 2015*; *Venturini et al., 2020*; *Lomakin et al., 2020*), choanoflagellates, the chytrid fungus *Batrachochytrium dendrobatidis* (*Fritz-Laylin et al., 2017b*) (although other chytrids, such as *Spizellomyces punctatus* can crawl in the absence of confinement [*Medina et al., 2020*]) and dictyostelid amoebozoans (*Srivastava et al., 2020*), suggesting that the ability to respond to confinement might be an ancient feature. These prior findings, together with our observation of an amoeboid switch in choanoflagellates, suggest that the switch from a flagellate to a crawling phenotype in response to confinement might have been part of an ancestral stress response in the last common choanozoan ancestor (*Brunet and Arendt, 2016*). The crawling behavior of the choanozoan ancestor might even have been more extensive than what we observe in modern choanoflagellates: indeed, most choanoflagellate species have secondarily lost some genes often involved in crawling motility, such as the integrin complex (*Richter et al., 2018*; *Helena et al., 2020*) and the transcription factors Brachyury (*Sebé-Pedrós et al., 2016*) and Runx (*Richter et al., 2018*; *Brunet and King, 2017*). Interestingly, among the choanoflagellate species we tested, the only one known to have retained Runx is *S. urceolata* (*Richter et al., 2018*), which displayed the most extensive crawling behavior under confinement (*Figure 6D*, *Video 15*). Another intriguing observation lies in the fact that mammalian leukocytes with artificially disrupted components of the integrin adhesome show little motility between two flat surfaces but retain the ability to migrate in 3D (*Lämmermann et al., 2008*) and to crawl through constrictions (*Reversat et al., 2020*) – reminiscent of our observation of the *S. rosetta* escape response.

Another open question is the nature of the biophysical mechanism that initiates bleb formation in choanoflagellates. In animal cells, bleb formation can result from one of the two possible mechanisms: breakage of the actomyosin cortex followed by detachment and blistering of the plasma membrane overlying the wound, or delamination of the plasma membrane from an initially intact actomyosin cortex (*Charras and Paluch, 2008*). These two mechanisms have sometimes proven

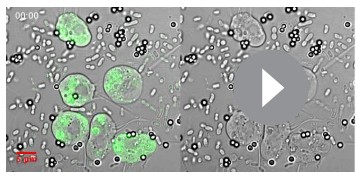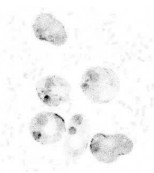

**Video 8.** Time-lapse of *S. rosetta* cells under 1 μm confinement expressing MRLC-mTFP (which marks myosin II) and imaged by confocal microscopy. Note the dynamic intracellular distribution of myosin II foci and fibers. Large fluorescent dots in the mTFP channel are autofluorescent food vacuoles previously described in *S. rosetta* (*Wetzel et al., 2018*).
https://elifesciences.org/articles/61037#video8

difficult to experimentally distinguish in animal cells (reviewed in *Paluch and Raz, 2013*), and it is currently unclear which underlies the initiation of blebbing in choanoflagellates.

Specialized crawling cell types are present in multiple animal lineages, including sponges (archeocytes), ctenophores (stellate cells), cnidarians (amoebocytes), invertebrate bilaterians (amoebocytes), and vertebrates (white blood cells and mesenchymal cells) (Table S1; *Figure 6—figure supplement 1*). Our results raise the possibility that that these might have evolved by stabilization of an ancestral stress response to confinement, representing an example of evolution of a cell type from temporally alternating phenotypes (in line with the 'temporal-to-spatial transition' hypothesis [*Mikhailov et al., 2009*; *Sebé-Pedrós et al., 2017*; *Brunet and King, 2017*]) and more specifically from a stress response (*Wagner et al., 2019*). In the single-celled ancestors of animals, confinement may have occurred in the context of interstitial media such as silts, or during attempts at phagocytosis by other predatory micro-eukaryotes. Interestingly, confinement activates blebbing in vertebrate mesenchymal and embryonic cells (*Liu et al., 2015*; *Ruprecht et al., 2015*; *Lomakin et al., 2020*; *Venturini et al., 2020*); in this case, the source of confinement in vivo is no longer the external environment, but is internal to the organism – for example, neighboring cells and/or dense extracellular matrix (*Lomakin et al., 2020*; *Venturini et al., 2020*). This suggests that, while the response to confinement might have been maintained during animal evolution, the predominant source of confinement might have switched from the external environment to the organism itself. Such 'internal pressure' might have existed at early stages in the evolution of multicellularity (*Jacobeen et al., 2018*): indeed, a recent study that used laser ablation to quantify the mechanical properties of multicellular rosettes in *S. rosetta* has shown that that cells within rosettes exert significant compressive stress onto each other (*Larson et al., 2020*). Future research will determine whether compressive stress might, in some conditions, be sufficient to activate amoeboid mobility within rosettes; intriguingly, individual cells are sometimes extruded from rosettes through poorly characterized mechanisms (*Dayel et al., 2011*) and 3D reconstructions of rosettes have revealed the presence of cells with pronounced cellular protrusions (*Laundon et al., 2019*). This suggests that the ability to respond to cell deformation – originally mobilized to escape external confined microenvironments – could have been co-opted during the evolution of multicellularity in response to stress exerted by neighboring cells. If so, both functions likely coexisted when multicellularity was facultative.

An open question is whether stimuli other than confinement can induce the amoeboid switch in choanoflagellates. The behavior of *S. rosetta* and other choanoflagellate has been extensively studied in response to a broad diversity of environmental stimuli, including multiple bacterial species (*Alegado et al., 2012*; *Ireland et al., 2020*; *Woznica et al., 2017*), pH (*Miño et al., 2017*), dissolved oxygen content (*Kirkegaard et al., 2016*), light (*Brunet et al., 2019*), and predatory heliozoans (*Kumler et al., 2020*), none of which has so far been observed to induce blebbing or crawling. While we cannot rule out the possibility that stimuli other than confinement might be able to induce amoeboid phenotypes (notably in species that have been studied less extensively than *S. rosetta*), it is currently unclear whether such other inducers exist.

If the switch between a flagellate phenotype and a crawling phenotype was fast and post-transcriptionally regulated in the last common

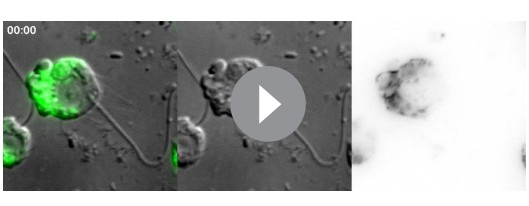

**Video 9.** Time-lapse of an *S. rosetta* amoeboflagellate cell under 1 μm confinement expressing MRLC-mTFP (which marks myosin II) and imaged by epifluorescence microscopy. Note that expanding blebs are devoid of myosin II and are re-invaded by myosin II before retraction, similar to F-actin.
https://elifesciences.org/articles/61037#video9

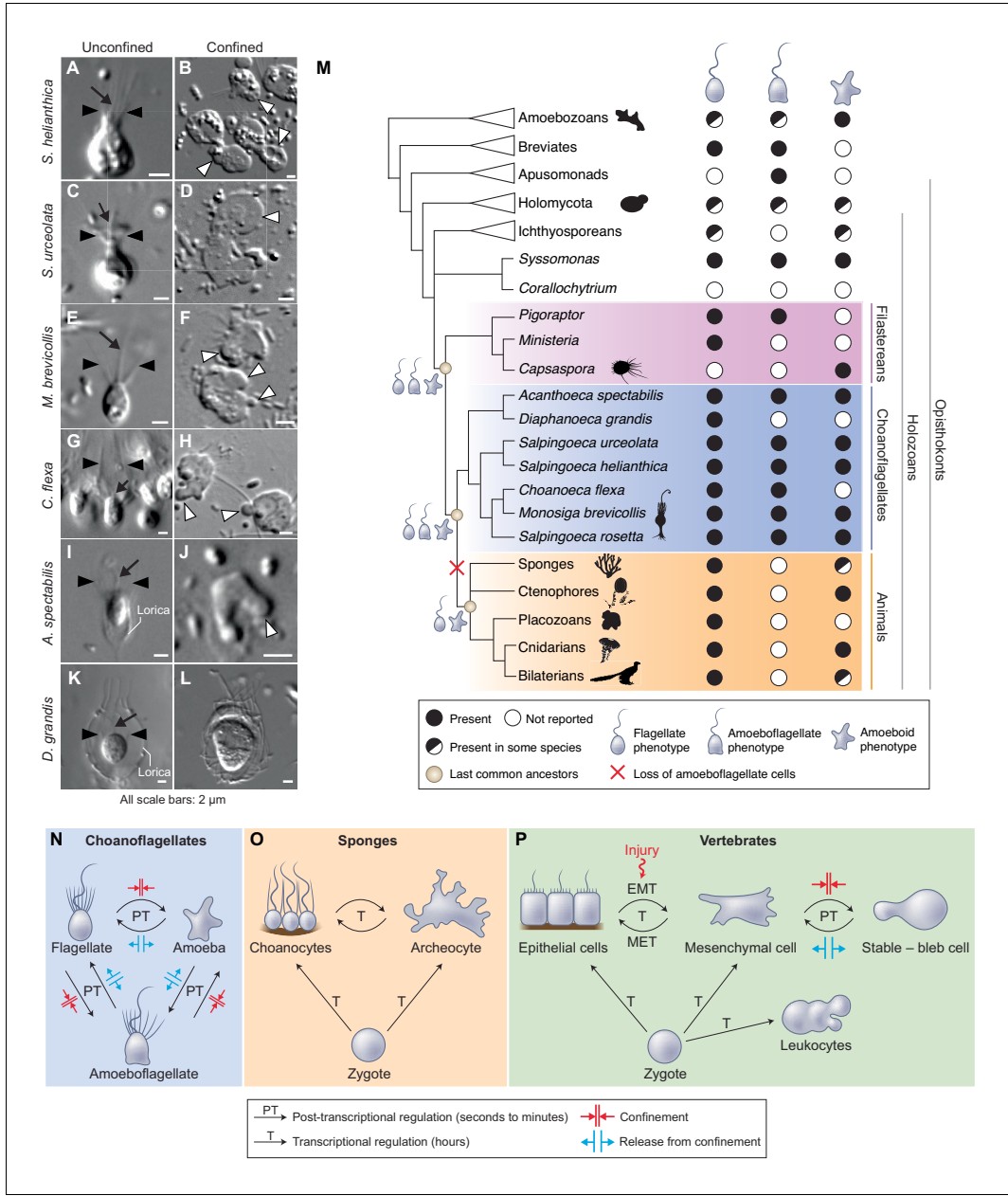

**Figure 6.** The last common choanozoan ancestor likely had amoeboid and flagellate life-history stages. (A–J) Five of six choanoflagellate species tested underwent the amoeboid transition under 2 μm confinement (***Videos 12–15***). (K–L) In contrast, the loricate choanoflagellate *Diaphanoeca grandis* was passively flattened under 2 μm confinement, but did not generate blebs (***Figure 6—video 1***). (M) Phylogenetic distribution of flagellate, amoeboflagellate, and amoeboid cell phenotypes in animals, fungi, amoebozoans, and their relatives. We infer that the last common ancestor of choanoflagellates and animals was able to differentiate into flagellate, amoeboid, and amoeboflagellate forms. Flagellate and amoeboid forms were likely both still present in the last common ancestor of all animals. See ***Figure 6—figure supplement 1*** and ***Supplementary file 1*** for full supporting evidence regarding the distribution of cellular phenotypes in animals. Species silhouettes are from Phylopic (http://phylopic.org). (N–P) Commonalities and differences in the regulation of cellular phenotypic transitions in choanoflagellates, sponges, and vertebrates. (N) Choanoflagellates can rapidly alternate (in a matter of minutes) between flagellate, amoeboflagellate, and amoeboid forms based on degree of external confinement (***Figure 1***). Inhibition of transcription does not prevent transitions between amoeboid and flagellate forms, suggesting that the transition is post-transcriptionally regulated (***Figure 6—figure supplement 2***). (O) In sponges, the zygote can give rise to flagellated choanocytes and amoeboid archeocytes. Choanocytes and archeocytes can reversibly interconvert, but this process takes several hours and likely requires transcriptional regulation

*Figure 6 continued on next page*

*Figure 6 continued*

(*Sogabe et al., 2019*). (P) In vertebrates, multicellular development from the zygote results in terminal differentiation of ciliated epithelial cells, mesenchymal cells and amoeboid leukocytes, but injury can trigger differentiation of epithelial cells into crawling mesenchymal cells (by epithelial-to-mesenchymal transition or EMT [*Lamouille et al., 2014*; *Dongre and Weinberg, 2019*]) that respond to confinement by switching to a stable-bleb form capable of amoeboid migration (*Liu et al., 2015*; *Ruprecht et al., 2015*). The switch from epithelial to mesenchymal cells is reversible (by mesenchymal-to-epithelial transition or MET).

The online version of this article includes the following video and figure supplement(s) for figure 6:

**Figure supplement 1.** Phylogenetic distribution of crawling cells, epithelial cells, collar cells, and flagellated sperm cells in animals.

**Figure supplement 2.** The amoeboid switch is not affected by transcription inhibition.

**Figure 6—video 1.** Time-lapse of a 2 μm-confined choanoflagellate of the species *Diaphanoeca grandis*.

https://elifesciences.org/articles/61037#fig6video1

choanozoan ancestor – as it is in modern choanoflagellates (*Figure 6N*; *Figure 6—figure supplement 2*) – it must have come under the control of transcriptional regulators during early animal evolution (*Lamouille et al., 2014*; *Figure 6O–P*). Alternatively, the alternation between amoeboid and flagellate forms might have been ancestrally transcriptionally regulated, possibly by other signals than confinement, and have evolved toward a 'streamlined' post-transcriptional stress response in choanoflagellates. Study of additional holozoans will be crucial in distinguishing between these hypotheses. Regardless of whether the switch between amoeboid and flagellate phenotypes was ancestrally transcriptional or post-transcriptional, it is likely that it was reversible in stem animals as it still is in sponges (*Sogabe et al., 2019*; *Figure 6O*), but later become irreversible with the evolution of terminal cell differentiation (*Figure 6P*).

Whether cell crawling was transient or constitutive in stem animals remains an open question. Although constitutively crawling cell types are present in most animals, they seem absent from calcaronean sponges (*Adamska, 2016*), placozoans (*Smith et al., 2014*), and xenacoelomorph worms (*Chiodin et al., 2013*). In these lineages, cell crawling often exists instead as a transient phenomenon. Indeed, in many animal lineages (including those with stable amoeboid cell types), transient cell crawling often contributes to embryonic development (for example, primordial germ cells often display amoeboid migration [*Grimaldi and Raz, 2020*; *Barton et al., 2016*; *Supplementary file 1*]) and/or to wound healing (that often involves crawling by cell types that do not normally display it, such as epithelial cells [*Lamouille et al., 2014*; *Supplementary file 1* and *Figure 6—figure supplement 1*]).

Another open question is the nature of the mechanotransduction pathway by which choanoflagellates detect and respond to confinement. Recently, vertebrate cells have been found to detect confinement at the level of the nucleus through a pathway that involves calcium release as well as the nuclear phospholipase cPLA2. This mechanism is unlikely to be involved in the choanoflagellate amoeboid switch for three reasons: (1) partly confined choanoflagellates show cortical relocalization of myosin II only within the confined parts of the cell (*Figure 4E*), suggesting that confinement is detected locally rather than by a central sensor such as the nucleus; (2) calcium signaling appears dispensable for the amoeboid switch (*Figure 4—figure supplement 3*); and (3) no homolog of cPLA2 is detectable in sequenced choanoflagellate genomes and transcriptomes (*Richter et al., 2018*; *King et al., 2008*; *Fairclough et al., 2013*). Thus, the nature of the confinement-sensitive pathway in choanoflagellates remains unclear. In the future, elucidation of this pathway will be necessary to test the hypothesis of homology between the choanoflagellate amoeboid switch and animal cell type differentiation mechanisms.

Finally, future comparative work will benefit from deeper insights into the mechanisms of cell crawling in choanoflagellates and animals (*Fritz-Laylin, 2020*). Intriguingly, the amoeboid cell types of sponges (archeocytes) express more genes shared with choanoflagellates than other sponge cell types do (*Sogabe et al., 2019*) (including choanocytes), which might include genes involved in crawling motility. Functional characterization of those genes in multiple phylogenetically relevant species as well as large-scale efforts to map animal cell type diversity within a phylogenetic framework (*Sebé-Pedrós et al., 2018*; *Arendt et al., 2016*; *Booth and King, 2020*) will help reveal how animal cell phenotypes have originated and evolved.

# Materials and methods

## Key resources table

| Reagent type (species) or resource | Designation | Source or reference | Identifiers | Additional information |
|---|---|---|---|---|
| Gene (*Salpingoeca rosetta*) | Regulatory myosin light chain short version (PTSG_00375) | NA | NCBI XM_004998867.1 | |
| Strain, strain background (*Salpingoeca rosetta*) | *S. rosetta* | PMID:24139741 | ATCC PRA-390; accession number SRX365844 | |
| Strain, strain background (*Algoriphagus machipongonensis*) | *A. machipongonensis* | PMID:22368173 | ATCC BAA-2233 | |
| Strain, strain background (*Echinicola pacifica*) | *E. pacifica* | PMID:16627637 | DSM 19836 | |
| Transfected construct (*S. rosetta*) | pEFl5'-Actin3'::pac-P2A-mTFP | *Wetzel et al., 2018* | Addgene ID NK676 | |
| Transfected construct (*S. rosetta*) | pEFL5'-Actin3'::pac, pActin5'-EFL3'::mCherry | This paper | Addgene ID NK802 | |
| Transfected construct (*S. rosetta*) | pEFL5'-Actin3'::pac, pActin5'-EFL3'::LifeAct-mCherry | This paper | Addgene ID NK803 | |
| Transfected construct (*S. rosetta*) | pEFL5'-Actin3'::pac, pActin5'-EFL3'::MRLC-mTFP | This paper | Addgene ID NK804 | |

## Choanoflagellate cultures

### Cultures of *S. rosetta*

*Salpingoeca rosetta* in the chain/slow swimmer form (*Dayel et al., 2011*) was maintained as a co-culture with the prey bacterium *Echinicola pacifica* (SrEpac) in 5% Sea Water Complete (SWC) culture medium, as previously described (*Levin and King, 2013*). Thecate *S. rosetta* were from a thecate SrEpac strain (HD1), which was produced from SrEpac through starvation following a published protocol (*Levin and King, 2013*). Rosettes were obtained from a co-culture of *S. rosetta* with the multicellularity-inducing bacterium *Algoriphagus machipongonensis* (*Dayel et al., 2011*; *Alegado et al., 2012*) (strain Px1) in 5% Cereal Grass Medium (CGM3) in Artificial Sea Water (ASW) (*King et al., 2009a*).

### Cultures of *C. flexa, S. helianthica, S. urceolata, D. grandis, M. brevicollis,* and *A. spectabilis*

Cultures of *S. urceolata, D. grandis, M. brevicollis,* and *A. spectabilis* were established by thawing frozen stocks stored in liquid nitrogen following a published protocol (*King et al.,*

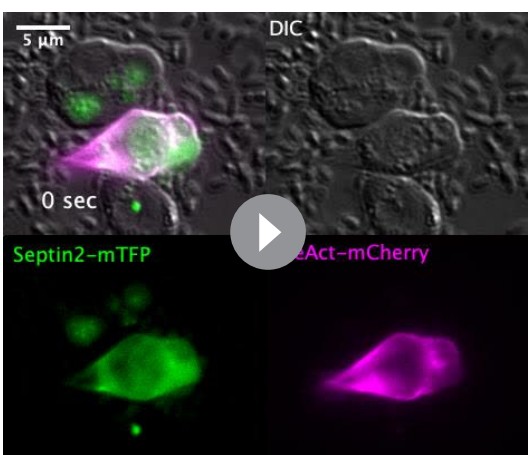

**Video 10.** Time-lapse of a crawling *S. rosetta* cell transfected with LifeAct-mCherry and septin2-mTFP. Note dynamic distribution of F-actin within the leading bleb (*Figure 5G–H*).
https://elifesciences.org/articles/61037#video10

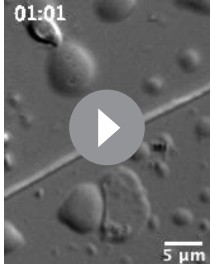

**Video 11.** Time-lapse of an escaping cell (from a similar experiment to the one shown in *Figure 5—video 1*) shedding two blebs into the unconfined space prior to escaping. One of the blebs is then reabsorbed by the cell during escape from confinement.
https://elifesciences.org/articles/61037#video11

*2009b*). Recipes for the culture media were previously published (*Richter et al., 2018*; *King et al., 2009a*) and modified as follows: *S. urceolata* was grown in 1% CGM3 at 25°C, *D. grandis* was grown in 5% CGM3 at 16°C, and *A. spectabilis* was grown at 16°C. *C. flexa* was obtained from a culture founded by a colony isolated from Curaçao in 2018 and continuously passaged since then as previously published (*Brunet et al., 2019*). Live cultures of *S. helianthica* were a gift from Mimi Koehl and Michael O'Toole II and were maintained in 25% freshwater CGM3 (FCGM3) following a published protocol (*Richter et al., 2018*).

## Live imaging

Cells were imaged by differential interference contrast (DIC) or epifluorescence microscopy using a 40× (water immersion, C-Apochromat, 1.1 NA), 63× (oil immersion, Plan-Apochromat, 1.4 NA), or 100× (oil immersion, Plan-Apochromat, 1.4 NA) Zeiss objective mounted on a Zeiss Observer Z.1 with a Hamamatsu Orca Flash 4.0 V2 CMOS camera (C11440-22CU).

## Confocal imaging

Fixed and stained samples were imaged by confocal microscopy using a Zeiss LSM 880 AxioExaminer with Airyscan and a 63×, 1.4 NA C Apo oil immersion objective (Zeiss) and excitation provided by a 405, 488, 568, or 633 nm laser (Zeiss).

## Cell confinement

### Dynamic cell confiner

A one-well dynamic cell confiner (*Liu et al., 2015*; *Le Berre et al., 2014*) comprising an Elveflow Vacuum/Pressure Generator and an Elveflow AF1 DUAL–Vacuum/Pressure Controller was purchased from 4Dcell (Montreuil, France), together with suction cups and with 1 µm, 2 µm, 3 µm, 4 µm, and 5 µm confinement slides.

All confinement assays were realized on cells mounted in FluoroDishes (World Precision Instruments FD35-100) under a confinement slide and imaged on a Zeiss Observer Z.1 (see above). *S. rosetta* dynamic confinement experiments were performed using SrEpac cultures that were dense, but not starving (~10⁶ cells/mL). In assays aiming at visualizing both flagellar retraction and regeneration in the same cells (*Figure 1—figure supplement 1*, *Figure 1—video 2*), the cells were attached to the substrate by coating the FluoroDish with 0.1 mg/mL poly-D-lysine for 1 min (Sigma–Aldrich P6407-5MG and washed twice quickly with ASW) before mounting the cells. In assays aimed at investigating the possibility of crawling (*Figure 1—video 1* and escape assays), poly-D-lysine was omitted.

Confinement was applied by following provider's instructions, by gradually decreasing pressure from −3 kPa to −10 kPa with the vacuum/pressure controller. Confinement was released by gradually restoring pressure to −3 kPa.

Escape assays were realized following the same protocol as confinement assays, but by

**Video 12.** Time-lapse of a population of *Monosiga brevicollis* confined in a thin liquid film, displaying intense blebbing and crawling (or gliding) motility.
https://elifesciences.org/articles/61037#video12

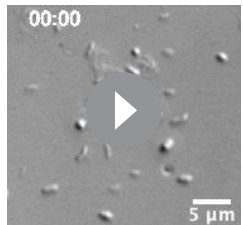

**Video 13.** Time-lapse of a 2 µm confined *Acanthoeca spectabilis* showing dynamic bleb extension.
https://elifesciences.org/articles/61037#video13

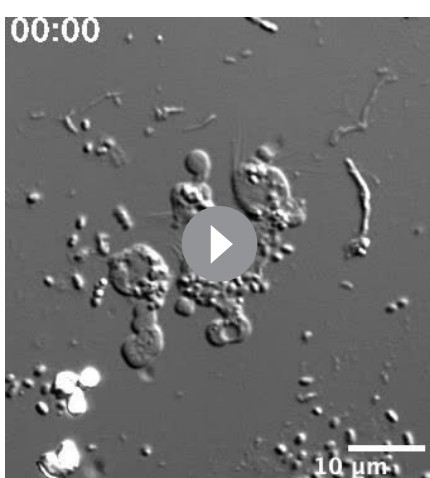

**Video 14.** Time-lapse of four *Salpingoeca helianthica* cells, 2 μm confined, displaying long dynamic blebs. https://elifesciences.org/articles/61037#video14

imaging, the cells trapped under the micropillars of a 3 μm confinement slide.

## Confinement with microbeads

Some early confinement experiments and pharmacological assays were performed by confining *S. rosetta* cell suspensions between two coverslips separated by microbeads (acting as spacers) and imaging them on a Zeiss Z.1 observer (see above).

The following types of microbeads were used: non-fluorescent 1 μm sulfate/latex beads (ThermoFisher Scientific S37498), non-fluorescent 2 μm sulfate/latex beads (ThermoFisher Scientific S37500), orange fluorescent 2 μm microbeads (Sigma–Aldrich L9529-1ML), and orange fluorescent 1 μm microbeads (Sigma–Aldrich L9654-1ML). Beads were stored at 4°C and resuspended in ASW prior to experimentation by centrifugation for 10 min at 10,000 g on a tabletop microcentrifuge followed by supernatant removal and resuspension.

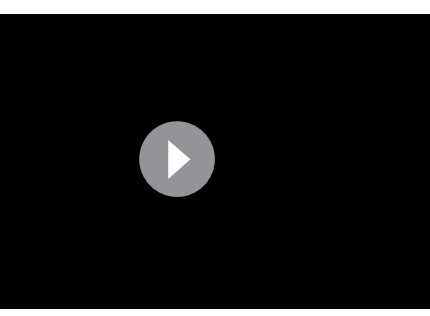

**Video 16.** Automated recognition of blebs in 2 μm-confined *S. rosetta* cells. Left panel: DIC channel. Middle panel: result of the cell segmentation. Right panel: cell protrusions, classified into expanding blebs (orange) and retracting blebs (blue). https://elifesciences.org/articles/61037#video16

**Video 15.** Time-lapse of a 2 μm confined *S. urceolata* cell crawling out of its theca. The cell crawls over about 40 μm, shedding cellular material at its rear end (possibly similar to the shedding of blebs by *S. rosetta*; *Video 11*). https://elifesciences.org/articles/61037#video15

Prior to confinement, 100 mL of a dense *S. rosetta* culture (strain SrEpac,~$10^6$ cells/mL) was filtered through a 5 μm syringe-top filter (Fisher Scientific SLSV025LS; to remove large biofilm pieces) and Percoll-purified (to remove bacteria) following a published protocol (*Levin and King, 2013*). The resulting *S. rosetta* suspension was further concentrated into 100 μL by centrifugation at 5000 g for 5 min on a tabletop microcentrifuge, thus reaching a final density of ~$10^9$ cells/mL. The resulting dense cell suspension was placed on ice and immediately mixed 10:1 with a stock suspension of microbeads in ASW. 0.1 μL of the cells/beads mixture was mounted on a rectangular coverslip pre-treated with a Corona surface treater (Electro-Technic Products BD-20AC) to facilitate liquid spreading and

surmounted with a second (non-Corona-treated) coverslip.

## Confinement in a thinly spread liquid layer

The first confinement experiments (*Video 3*) were realizing by trapping cells into a thinly spread layer of liquid medium surmounted by oxygen-permeant oil. Five microliters of a dense cell suspension (concentrated down to ~$10^9$ cells/mL in ASW complemented with 1% CGM3% and 1% rhodamine-dextran as a fluorescent marker of the aqueous phase [Sigma–Aldrich D6001]) were spread on a FluoroDish pre-treated with a handheld Corona surface treater and surmounted with 120 µL of oxygen-permeant anti-evaporation oil (Ibidi 50051). The thickness of the medium layer was measured by visualizing rhodamine-dextran fluorescence using a confocal microscope. Rhodamine-dextran fluorescence was exclusively observed within the aqueous layer of ASW-based medium (containing the cells) and was excluded from the overlaying oil (consistent with rhodamine-dextran being hydrophilic). Confocal stacks were visualized with Fiji (*Schindelin et al., 2012*) and the decrease of red fluorescence at the water/oil interface allowed quantification of the thickness of the aqueous phase. Film thickness varied within a given FluoroDish (possibly due to meniscus effects) and ranged from 1 to 8 µm. Cells were observed to be consistently amoeboid if they were trapped in a layer thinner than 3 µm and to be consistently flagellate and free-swimming if the layer was thicker than 5 µm – consistent with observations made with confinement slides and microbeads.

## Transfection

For F-actin and septin2-mTFP live imaging (*Video 10*), cells were co-transfected with plasmids encoding LifeAct-mCherry (Addgene ID NK612) and septin2-mTFP (Addgene ID NK641, which distributes within the entire cytoplasm in highly-expressing cells) following a published transfection protocol (*Booth et al., 2018*) and imaged in epifluorescence microscopy using a Zeiss Z.1 observer.

Stable mTFP-expressing *S. rosetta* were produced following a previously published protocol (*Wetzel et al., 2018*) by transfection (*Booth et al., 2018*) of a plasmid encoding a puromycin resistance protein followed by mTFP, separated by a P2A self-cleaving peptide (Addgene ID NK676).

Initial attempts at producing stable LifeAct-mCherry and MRLC-mTFP-expressing *S. rosetta* using the P2A system (similar to the system for mTFP expression) failed to generate any detectable puromycin-resistant cells after 10 days of puromycin selection. Thus, an alternative strategy was devised in which the fluorescent marker of interest and the Pac puromycin resistance protein were expressed as two distinct ORFs under the control of two distinct promoters within the same plasmid. To obtain an initial proof of principle, plasmids were designed which contained a first open reading frame (ORF) with the Pac gene fused to the 5′-UTR and promoter of *S. rosetta* EFL and the 3′-UTR of *S. rosetta* actin (5′EFL-Pac-3′Actin), and a second ORF with the mCherry gene fused to the promoter and 5′-UTR of *S. rosetta* actin and the 3′-UTR of *S. rosetta* EFL (5′Actin-mCherry-3′EFL). A fragment containing the EFL 5′-UTR and Pac gene and a fragment containing the Actin 3′-UTR were amplified from Addgene ID NK676 (as two separate amplifications) and then assembled together in pUC19 digested with BamHI in a Gibson assembly reaction (NEB HiFi). This 5′EFL-Pac-3′Actin fragment was then amplified and inserted into Addgene ID NK648 (a plasmid that contained the 5′Actin-mCherry-3′EFL construct) cut with either XbaI or HindIII. This yielded constructs with all four possible orders and relative orientations of 5′EFL-Pac-3′Actin and 5′Actin-mCherry-3′EFL. All four constructs were tested in eight transfection reactions each. After 7 days of puromycin selection, one transfection reaction with one of the four constructs was observed to have given rise to red mCherry-expressing cells whose fluorescence was stable across at least three cell passages. This construct was deposited on Addgene with the Addgene ID NK802.

The NK802 plasmid was then used as a backbone for insertion of LifeAct-mCherry and MRLC-mTFP. LifeAct-mCherry was extracted from a pre-existing backbone by restriction digestion with AflII and NcoI (Addgene ID NK612) and inserted in the backbone of NK802 by restriction cloning. The resulting Life-Act-mCherry/Pac plasmid was deposited on Addgene with the ID NK803. The full ORF of Sr-MRLC was cloned from *S. rosetta* cDNA as in *Booth et al., 2018* and fused with mTFP through insertion into the NK676 backbone by In-Fusion cloning (Takara), separated by the linker sequence DYKEPVAT (nucleotide sequence GACTACAAGGAACCGGTCGCCACC, following the *Drosophila melanogaster* MRLC fusion construct *sqh-gfp* [*Royou et al., 2002*]). MRLC-mTFP was extracted from the resulting plasmid by digestion with AflII and NcoI and inserted into the NK802

backbone by restriction cloning. The resulting MRLC-mTFP/Pac plasmid was deposited on Addgene with the ID NK804.

Either 5 or 10 µg of NK803 and NK804 was tested in 24 transfection reactions each (resulting in 96 reactions in total) as in *Booth et al., 2018* and selection with 80 µg/mL puromycin was performed as in *Wetzel et al., 2018*. Resistant cells were observed after 7 days in five reactions with 5 µg of NK803 and in 13 reactions with either 5 or 10 µg of NK804. For each construct, three reactions with visible mCherry or mTFP fluorescence were chosen for passaging and diplayed stable fluorescence and puromycin resistance over at least three passages, suggesting either genomic integration of the transfected construct or stable replication as an episomal element. Fluorescence patterns appeared identical in all three strains for each construct.

## Cell segmentation and morphometrics

For escape response assays (*Figure 5*), cell shapes were segmented using the DIC channel. Cell segmentations were obtained from the predictions of a StarDist model (*Schmidt et al., 2018*; *Weigert et al., 2020*). Ground truth for training the StarDist model was created by cropping out and manually labeling a subset of the cells to be analyzed, at evenly distributed time points throughout the movies. A total of N = 160 square images (width 151px) of individual cells and their associated masks were rearranged into 32 mosaic images containing 5 × 5 cell images, representing the ground truth for network training. The centers of the segmented cells were tracked over time using Trackpy (*Allan et al., 2019*).

Two distinct movies were analyzed, which led to 94 cells being segmented, of which eight escaped. Cell centroid, aspect ratio, projected surface area, and circularity were computed from the segmented shapes. The confinement boundary was manually drawn as a circle at the position of the pillar border, which allowed computation of the distance between the boundary and the cell centroid, the cell front (defined as the minimal distance between the border and all points within the cell), and the cell rear (defined as the maximal distance between the border and all points within the cell).

For calcium depletion assays (*Figure 4—figure supplement 3*), as no strong difference in blebbing activity was readily observable between the different conditions, we set up a pipeline for automated quantification of blebbing activity to test for possible small, quantitative differences. The analysis was performed by M.A. who was blinded to the treatment conditions. Cells were segmented using the DIC channel. A StarDist model was trained using 14 manually labeled movie crops as ground truth (size 1024 × 1024 px), containing comparable amounts of cells for each of the four different perturbations to be analyzed. Prior to prediction, the image background was estimated by applying a gaussian filter (kernel size 20 px) and substracted from the input images. To improve the accuracy of the cell boundary reconstructions, the StarDist output was further processed by using the centers of the star-convex object predictions as seeds for a watershed segmentation based on the thresholded pixel predictions produced by the StarDist model's UNet (*Ronneberger et al., 2015*).

Blebbing activity was approximated as the rate of cell shape change. First, all movies were resampled to a frame rate of 1 frame per 20 s and only the first 240 s were considered. To compensate for a possible global drift of the field of view, all resulting frames of each movie were first registered to the first frame. For each cell and for each time point, the tracked cell labels were used to extract the difference between the shape of the cell and the shape of the same cell 20 s earlier. The zones over which cell shape differed between both time points matched recognizable blebs (see *Video 16* for an example). Blebbing activity for an individual cell was calculated as the mean area of this shape difference, averaged over all time points, and normalized by cell area. Cell tracks shorter than 200 s and those meeting any of the following conditions for any considered time point were excluded from this analysis:

- Cell area < 500 px (pixel spacing: 0.1625 µm)
- Cell displacement between time points > 5px
- Cell area change between time points > 10%
- Cell circularity < 0.7

Cell segmentation, tracking, and all downstream analysis were performed in Python (3.7) in combination with software belonging to the SciPy ecosystem (RRID:SCR_008058 [*Oliphant, 2007*;

*van der Walt et al., 2011*; *Hunter, 2007*; *van der Walt et al., 2014*]) and additional software Czi-File: https://pypi.org/project/czifile/ and Fiji (RRID:SCR_002285 [*Schindelin et al., 2012*]). This analysis concluded that blebbing activity did not differ significantly between the four conditions tested (*Figure 4—figure supplement 3*).

## Pharmacological assays

For all pharmacological inhibition assays, cells were pre-treated with small molecule inhibitors for 30 min before imaging. Negative controls were treated for the same time with a concentration of compound vector (most often DMSO) equivalent to that used in the highest inhibitor dosage. Compounds used and their stock and working concentrations are in *Supplementary file 2*.

Technical replicates were defined as individual cells. Biological replicate were batches of cells, treated, and processed together. There were at least three biological replicates per condition, with at least five cells (technical replicates) per biological replicate. In each experiment, biological replicates were produced by splitting a single batch of cells into groups that were processed identically, except for the experimental variable of interest. Sample sizes can be found in the legend of each relevant figure. No outliers were encountered, and no data were excluded.

## Calcium depletion

For depletion of external calcium, cells were transferred into calcium-free AK sea water (CF-AKSW). CF-AKSW was prepared following a published AKSW recipe (*Booth et al., 2018*), omitting only $CaCl_2$, and further adding 20 mM EGTA to chelate any remaining calcium. A suspension of ~$10^8$ *S. rosetta* cells was concentrated into 100 µL (see 'Cell confinement – Confinement with microbeads' section) and resuspended in 1 mL CF-ASKW. Cells were then washed three times in 1 mL CF-AKSW in 1.5 mL plastic tubes by centrifugation ($2\times$ 5 min and $1\times$ 15 min) at 10,000 g in a tabletop microcentrifuge. Cells were then confined using 2 µm microbeads as spacers as detailed above. Microbeads were similarly resuspended in CF-AKSW before being added to the cells.

For depletion of intracellular calcium, cells were incubated with 327 µM BAPTA-AM (a cell-permeant calcium chelator) for 30 min before confinement and imaging.

## Immunostainings of flagellate and amoeboid cells

Immunostainings of flagellated *S. rosetta* cells were performed following a previously published protocol (*Booth et al., 2018*). Immunostainings of confined cells were performed by mounting cells between a small square coverslip (18 × 18 mm, VWR 470019–002) and a larger rectangular coverslip (24 × 50 mm, VWR 48393–241) using 1 µm or 2 µm microbeads as spacers (see 'Cell confinement – Confinement with microbeads'). Immediately after confinement, the two lateral sides of the small coverslip (parallel to the long side of the large rectangular coverslip) were glued to those of the large coverslip using a small quantity of Super Glue gently spread with a micropipette tip (*Figure 2—figure supplement 2*). This maintained close apposition of the two coverslips (and thus cell confinement) during the following steps. The Super Glue was left to dry for 5 min. This defined a flow chamber in which fixation, staining, and washing solutions could be pipetted on top of the large coverslip, close to the non-glued edges of the small coverslip, and then spread in the confined space by capillary action (*Figure 2—figure supplement 2*).

Fifty microliters of fixation solution (4% paraformaldehyde [PFA] in cytoskeleton buffer [*Booth et al., 2018*]) was pipetted close to the inflow side of the flow chamber (top in the last panel of *Figure 2—figure supplement 2*). The slide was transferred into a small parafilm-sealed box, in which air humidity was maintained by water-soaked paper towels (to prevent evaporation of liquid in the sample). Fixation was left to proceed for 2 hr at room temperature. After each incubation step, excess liquid was removed by absorbing it with a Kimwipe (Fisher Scientific 06–666) on the outflow side (bottom in the last panel of *Figure 2—figure supplement 2*), and the following solution was immediately added to the inflow side. After fixation, all washing steps were performed as when immunostaining flagellated cells, with the single difference that incubation in the primary antibody solution was allowed to proceed overnight at 4°C.

The primary anti-tubulin antibody was rat anti-α-tubulin (YOL1/34, Abcam ab6161, RRID:AB_305329, diluted 1:300). The secondary antibody was Alexa 647-anti-rat (ThermoFisher Scientific A-21247, RRID:AB_141778) diluted to the providers' specifications. F-actin was stained with 0.66

units/mL Alexa 647-phalloidin (Life Technologies A22287) when combined with FM 1–43 FX, or 0.66 units/mL rhodamine-phalloidin (Life Technologies R415) when combined with myosin II immunostaining. DNA was stained with 10 µg/mL Hoechst 3342 (ThermoFisher Scientific H3570).

For co-staining with FM 1–43 FX (which stains the plasma membrane in live cells and redistributed to the cytoplasm in fixed cells) and fluorescent phalloidin, 5 µg/mL FM 1–43 FX (ThermoFisher Scientific F35355, dissolved in water as single-use aliquots) and 0.66 units/mL Alexa 647-phalloidin were directly included in the fixation solution (together with 4% PFA and 6% acetone) in order to minimize the number of necessary washing steps. Fixation and staining were left to proceed for 2 hr at room temperature. The fixation/staining solution was washed once with PEM buffer (*Booth et al., 2018*) and once with 70% glycerol/PEM (with 5 mg/mL 1,4-diazabicyclo[2.2.2]octane [DABCO, Sigma–Aldrich D27802-100G] as an anti-fading agent) for mounting prior to confocal imaging.

## Acknowledgements

We thank D Booth, F Leon, M Sigg, and R Aldayafleh for help with *S. rosetta* transfections; D Booth and A Mulligan for cloning advice and assistance; the staff and students of the 2018 Physiology course in the Marine Laboratory (Woods Hole, MA); A Jha, O Dudin, T Linden, D Richter, D Booth, and F Rutaganira for feedback on the manuscript; V Ruprecht and L Fritz-Laylin for feedback on the project; D Maizels for assistance with the figures; and members of the King lab for stimulating discussions. TB was supported by the EMBO long-term fellowship (ALTF 1474–2016) and by the Human Frontier Science Program long-term fellowship (000053/2017 L).

## Additional information

### Funding

| Funder | Grant reference number | Author |
| --- | --- | --- |
| Howard Hughes Medical Institute | | Nicole King |
| EMBO | ALTF 1474-2016 | Thibaut Brunet |
| Human Frontier Science Program | 000053/2017-L | Thibaut Brunet |

The funders had no role in study design, data collection and interpretation, or the decision to submit the work for publication.

### Author contributions

Thibaut Brunet, Conceptualization, Resources, Formal analysis, Funding acquisition, Investigation, Methodology, Writing - original draft, Writing - review and editing; Marvin Albert, Software, Investigation, Visualization, Methodology; William Roman, Investigation, Visualization; Maxwell C Coyle, Resources, Methodology; Danielle C Spitzer, Investigation; Nicole King, Conceptualization, Resources, Supervision, Funding acquisition, Writing - original draft, Project administration, Writing - review and editing

### Author ORCIDs

Thibaut Brunet [iD] https://orcid.org/0000-0002-1843-1613
Danielle C Spitzer [iD] http://orcid.org/0000-0003-4827-1857
Nicole King [iD] https://orcid.org/0000-0002-6409-1111

### Decision letter and Author response

Decision letter https://doi.org/10.7554/eLife.61037.sa1
Author response https://doi.org/10.7554/eLife.61037.sa2

## Additional files

**Supplementary files**

- Supplementary file 1. Crawling cells across animal diversity.
- Supplementary file 2. Pharmacological compounds used in inhibitor assays.
- Transparent reporting form

**Data availability**

All data generated or analysed during this study are included in the manuscript and supporting files.

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
