## [Decision Letter]

**Acceptance summary:**

This paper elegantly describes, using cell biological approaches, that choanoflagellates (the sister-group to animals) can move using blebbing migration. Choanoflagellate cells can use this blebbing motility to escape confinement. This work provides evidence for the hypothesis that blebbing motility predated the diversification of animals.

**Decision letter after peer review:**

Thank you for submitting your article "A flagellate-to-amoeboid switch in the closest living relatives of animals" for consideration by *eLife*. Your article has been reviewed by two peer reviewers, and the evaluation has been overseen by a Reviewing Editor and Patricia Wittkopp as the Senior Editor. The following individual involved in review of your submission has agreed to reveal their identity: Purificación López-García (Reviewer #2).

The reviewers have discussed the reviews with one another and the Reviewing Editor has drafted this decision to help you prepare a revised submission.

Summary:

In this paper, Brunet and co-authors show that various species of choanoflagellates have the capacity to switch from the typical flagellated stage to an amoeboid, non-flagellated stage. They show these amoeboid cells move using blebbing migration. The paper makes four major points:

1) Several species of choanoflagellates make dynamic protrusions that appear similar to blebs in DIC when confined. This claim is well supported by nice quantitative analysis of blebbing using a diversity of choanoflagellate species.

2) The blebs made by choanoflagellates, like those of animal cells and *Dictyostelium*, involve breakage and healing of the actomyosin cortex. This is the weakest part of the paper (see below for further details).

3) Choanoflagellate cells can use this blebbing motility to escape the confinement, a concept supported by videos of cells doing this.

4) Amoeboid motility is homologous in choanoflagellates and animals. In particular, the authors postulate that epithelial and crawling cells in animals differentiated by exploiting the switch from the flagellated to amoeboid stages that existed in unicellular opisthokonts. Although this is an interesting hypothesis, we believe the data is not conclusive and both the implications and the conclusions need to be better explained in the general context of eukaryotic evolution.

Below we define what we believe should be done to improve the manuscript

Essential revisions:

1) The idea that amoeboid morphology and blebbing motility is older than animals is not particularly controversial: blebbing has been documented in various microbial lineages for some time, and blebbing motility uses a contractile actin cortex, which are also widely distributed. This, combined with a lack of engagement with alternative hypotheses weakens the conceptual significance of the paper. Fleshing out what the alternative hypotheses are, and/or providing context for how this work provides new insight into the evolution of blebbing would improve the paper. Furthermore, the Introduction should include more background information to distinguish blebbing motility from actin pseudopod based motility. Also, the use of the term "actin-filled" to discuss bleb retraction is confusing; do the authors mean actin-encased? Also a clear definition of what a "retracted bleb" is should be provided. In general, some information in the Results that could be better incorporated in the Introduction as it explains the authors' motivation for the work.

2) Similarly, the idea of homology between the amoeboid cells of choanoflagellates and animal amoeboid cells seems not well supported or at least no more than a potential homology of many eukaryotic amoeboid cells. This needs to be toned-down and/or discussed into the context of the potential ancestral eukaryotic feature. The same with the related idea that epithelial and crawling cells in animals differentiated by exploiting the switch from the flagellated to amoeboid stages that existed in unicellular opisthokonts.

Finally, the authors say that the switch between flagellated and crawling cells in choanoflagellates is triggered by particular size-related stress. However, it is difficult to imagine that animals evolved under such a type of stress. It may be interesting to discuss whether the authors have tried to see if alternative sources of stress induce transition to amoeboid states or, alternatively, discuss particular hypotheses about which kinds of stress might trigger this response. When discussing this, it may be worth considering an alternative: that choanoflagellates might be a side phylogenetic group having evolved specific characteristics and virtually lost amoeboid stages except for extreme situations like the ones shown here. The ancestor of metazoans would then had simply retained an ancestral (eukaryotic) capability to transition from flagellated to amoeboid states during the opisthokont life cycle without this capability being in any way related to volumetric stress but rather to particular environmental clues.

Overall, all these ideas should be discussed in the context of eukaryotic evolution and toning down the potential implications.

3) The authors use the standard definition of blebbing: actomyosin cortex breakage and healing concomitant with production of round protrusions devoid of actin. The paper provides insufficient data to support the claim that choanoflagellate cells defined as "amoeboid" based on the lack of microvilli undergo this form of blebbing. Providing additional examples and/or quantitative analyses of the data would strengthen the paper. Specifically:

a) Only a single cell with LifeAct is shown (Figure 2C, with what looks to be the same cell in Video 7). Additional examples would be welcome. However, this cell has high levels of septin overexpression: could this be interfering with the native phenotype? Figure 3K looks very different from Figure 2Q. Is septin localizing to the membrane? The WT cells shown in P do not show cortical actin. It seems likely worthwhile repeating this experiment with LifeAct but without septin overexpression. Additionally, linescans to quantitate the cortical actin levels before, during, and after cortical breakage would provide quantitative support, particularly with a membrane stain, or at the very least, the septin localization.

b) The phalloidin staining in Figure 2 and Figure 2—figure supplement 1 is not particularly convincing in terms of the presence of cortical actin. Why do the cells in Figure 2P and W look different? The additional examples also show weak actin staining.

c) Figure 2S and V show myosin staining which does not appear to be localized to the cortex. More than a single cell should be shown, along with quantitation by line scan analysis to support the claim of cortical localization.

d) Definition of amoeboid seems problematic as many of the images show "amoeboid" cells with what look to be microvilli: 2C-K, 2U-W, 3A-F

e) The text describing the phalloidin staining is a bit circular as it assumes blebbing to interpret the staining patterns, but then uses the staining pattern as confirmation of blebbing.

4) The authors indicate that the actin in amoeboid choanoflagellates undergo retrograde flow. The authors show a single cell imaged with widefield fluorescence (Figure 3F-G, Video 9). Typically retrograde flow is visualized using TIRF microscopy to show the movement of individual actin filaments within a network. The data presented makes it difficult to evaluate whether this is the retraction of the entire cortical network, or flow of actin filaments within a network. To support a claim of retrograde flow, additional data and analysis should be provided. Moreover, the concept of retrograde flow in the context of blebbing motility needs to be explained more fully in the text.

5) The microtubule inhibition experiments involve treating cells for 36 hours with a non-standard microtubule inhibitor. Due to the possibility of off-target effects, the authors should repeat this experiment with a second microtubule inhibitor to cross-validate the result. A second, orthogonal approach would be to stain cells and look for anti-correlation between microtubule density and bleb formation.

---

## [Author Response]

Essential revisions:1) The idea that amoeboid morphology and blebbing motility is older than animals is not particularly controversial: blebbing has been documented in various microbial lineages for some time, and blebbing motility uses a contractile actin cortex, which are also widely distributed. This, combined with a lack of engagement with alternative hypotheses weakens the conceptual significance of the paper. Fleshing out what the alternative hypotheses are, and/or providing context for how this work provides new insight into the evolution of blebbing would improve the paper.

We agree that the idea that amoeboid motility (including blebbing motility) is older than animals had been proposed before our paper. Indeed, we supported this hypothesis ourselves in a review paper we wrote before we were aware of the existence of the amoeboid cells in *S. rosetta* (Brunet and King, 2017). As the reviewer states, that idea was consistent with the presence of blebbing in *Dictyostelium* and other amoebae. We now mention this explicitly in the Introduction:

“Consistent with a pre-metazoan origin of cell crawling, cell biological and biochemical studies have revealed similar cellular structures and conserved molecules involved in cellular crawling in animals and protists. […] The cellular and biochemical similarities among cellular protrusions from animals and protists are consistent with a shared origin (Fritz-Laylin, 2020).”

However, this idea has also been questioned by some authors, including a recent influential review on animal origins (Cavalier-Smith, 2017), which posited explicitly that animals evolved from “phagotrophic non-amoeboid flagellates” and that “amoebae are not primitive but arose from zooflagellate ancestors independently in each of the three ancestrally biciliate eukaryotic supergroups”. Without stating it explicitly, most other review papers on animal origins have depicted the single-celled ancestor of animals as a flagellate without other cell phenotypes, with amoeboid cells only being introduced after multicellularity evolved – thus suggesting the idea of independent evolution of amoeboid phenotypes (King, 2004; Nielsen, 2008). There have thus been a diversity of views on this question in the recent literature. Following the reviewers’ suggestion, we have now added a paragraph in the Introduction to explicitly present the breadth of alternative hypotheses regarding the ancestry of amoeboid cells:

“However, the discontinuous phylogenetic distribution of cellular protrusions involved in crawling motility has also been interpreted as evidence for convergent evolution of amoeboid cells (Cavalier-Smith, 2017), and genomes do not resolve the controversy because proteins involved in crawling motility can also fulfill crawling-independent functions. […] Thus, conservation in choanoflagellates of proteins required for crawling cell motility in animals is also consistent with an alternative hypothesis, in which their functions were restricted to locomotion-independent functions in the last common choanozoan ancestor and subsequently co-opted during the independent evolution of locomotory protrusions in animal crawling cells and amoeboid protists, as suggested by (Cavalier-Smith, 2017) and others.”

While we think this alternative hypothesis has become less likely in light of our results (which expand the known taxonomic distribution of blebbing and of crawling locomotion), it remains difficult to rule out, reflecting the difficulty of ascertaining the homology of phenotypes over large evolutionary distances. We have added a sentence to the conclusion to emphasize this remaining uncertainty.

“However, evolutionary convergence remains difficult to entirely rule out, given that the molecules known to be required for blebbing (such as actin and myosin II) have other functions. Future research in choanoflagellates, amoebozoans and animals might help determine whether specific regulators of blebbing exist (similar to the ancient eukaryotic “flagellar toolkit” (Carvalho-Santos et al., 2011)).”

Furthermore, the Introduction should include more background information to distinguish blebbing motility from actin pseudopod based motility.

In addition to the definitions of blebbing motility and pseudopod-based motility that were previously provided in the Results, we have now added expanded descriptions of blebbing motility and pseudopod-based motility to the Introduction (which are part of the paragraph copied above and are included again below):

“These include cellular protrusions frequently involved in locomotion: (1) filopodia, slender, finger-like protrusions containing bundles of actin filaments, that are found in animals cells, choanoflagellates and filastereans Sebé-Pedrós et al., 2013; (2) pseudopods, which contain branched F-actin networks reticulated by the Arp2/3 complex downstream of the actin regulators SCAR and WASP and are observed in mammalian neutrophils and chytrid fungi (Fritz-Laylin et al., 2017b); and (3) blebs, which form as F-actin-free protrusions by delamination of the actomyosin cortex from the plasma membrane under the influence of actin/myosin II cortex contractility. Blebs have been observed in animal cells (such as mammalian macrophages and zebrafish primordial germ cells) and in free-living amoebae such as *Entamoebahistolytica* and *Dictyosteliumdiscoideum* (reviewed in (Charras and Paluch, 2008; Paluch and Raz, 2013)).”

Also, the use of the term "actin-filled" to discuss bleb retraction is confusing; do the authors mean actin-encased?

We thank the reviewer for pointing this out. We do mean actin-encased and have corrected the text accordingly. For example, we state that, “we observed both F-actin-free and F-actin-encased protrusions, consistent with our observations of expanding and retracting blebs, respectively (Figure 2L-S; Figure 3).”

Also a clear definition of what a "retracted bleb" is should be provided. In general, some information in the Results that could be better incorporated in the Introduction as it explains the authors' motivation for the work.

Thank you for this suggestion. In the manuscript, we focus on dynamic processes and therefore favor “retracting” over “retracted” and “expanding” over “expanded.” We now provide explicit definitions for “retracting blebs” and “expanding blebs” in the legend of Figure 2:

“We observed that expanding blebs (e; defined as blebs that increased in size during the period of observation) were cytoplasm-filled, but F-actin-free. (E, H, K) We found that F-actin subsequently re-invaded blebs that then initiated retraction (r; defined as designating retracting blebs that decreased currently decreasing in size during the period of observation) were re-invaded by F-actin.”

As noted in response to a suggestion above, we have also added information on pseudopods, blebs, and their underlying mechanisms and phylogenetic distribution in the Introduction.

2) Similarly, the idea of homology between the amoeboid cells of choanoflagellates and animal amoeboid cells seems not well supported or at least no more than a potential homology of many eukaryotic amoeboid cells. This needs to be toned-down and/or discussed into the context of the potential ancestral eukaryotic feature.

Thank you for raising these points. If we can restate what we think might be your concern, we believe you are asking us to address why (and if) these results in choanoflagellates provide stronger evidence of a pre-metazoan ancestry of crawling motility and blebs.

As the reviewer states, crawling motility is known in a broad diversity of single-celled eukaryotes. Notably, detailed studies of amoebozoans (in particular *Dictyostelium* (Merkel et al., 2000; Yoshida and Soldati, 2006) and *Entamoeba* (Maugis et al., 2010)) have revealed the presence of blebs that rely on the same molecular and cellular mechanisms as those observed in animals (reviewed in Charras and Paluch, 2008; Paluch and Raz, 2013). While these findings are consistent with a possible pre-metazoan ancestry of blebs, the vast phylogenetic distance between amoebozoans and metazoans means that convergence remains a plausible interpretation in the absence of other supporting data (Cavalier-Smith, 2017). Indeed, while crawling motility has been observed in some intervening lineages (e.g. in ichthyosporeans and chytrid fungi), blebs have not yet been documented in these organisms.

We provide new discussion of these ideas here:

“Outside choanozoans, blebs have been well-described in amoebozoans, notably *Dictyostelium discoideum* (Merkel et al., 2000; Yoshida and Soldati, 2006) and *Entamoeba histolytica* (Maugis et al., 2010). Cell biological studies have revealed close structural and mechanistic similarities between amoebozoan and metazoan blebs, consistent with, but not demonstrative of, a pre-metazoan origin of blebbing.”

This leaves us with the question of whether similar data from choanoflagellates add meaningfully to our understanding of the ancestry of crawling motility and blebs in metazoans.

Importantly, the likelihood of homology between two structures is not only a function of their similarity, but also depends on their phylogenetic distribution (Pinna de, 1991; Remane, 1983; Wagner, 1989). Because choanoflagellates and metazoans are sister-groups, morphologically similar features that are also based on the same molecular machinery in these two groups (including blebs) are most parsimoniously interpreted as homologous. Therefore, we argue that our findings strengthen the inference that crawling motility and blebs predate the ancestry of metazoans. We now detail this reasoning explicitly in the main text:

“However, the absence of described blebs in intervening branches (fungi, ichthyosporeans, filastereans and choanoflagellates) raised the alternative possibility that blebs might have evolved convergently in amoebozoans and animals by independent co-option of cortical actomyosin contractility, which could have been ancestrally involved in other functions (such as cytokinesis; see Introduction). […] Our results thus provide novel evidence in favor of a pre-metazoan origin of blebbing and of crawling motility.”

However, we agree with the reviewer that convergence cannot be fully ruled out, notably in light of the broad phylogenetic distribution and functional promiscuity of proteins (F-actin and myosin II) known to be required for blebbing. We have added a sentence to be explicit about this limitation:

“However, evolutionary convergence remains difficult to entirely rule out, given that the molecules known to be required for blebbing (such as actin and myosin II) have other functions[…] Further comparative evidence might also come from investigation of the possibility of blebbing in other single-celled opisthokonts such as fungi, ichthyosporeans and filastereans, for example (but not necessarily only) in response to confinement.”

The same with the related idea that epithelial and crawling cells in animals differentiated by exploiting the switch from the flagellated to amoeboid stages that existed in unicellular opisthokonts.

Thank you for raising this point. We proposed this hypothesis because it is consistent with the existing data and testable, although we acknowledge there are other possible explanations for evolution of cell differentiation in animals. One key missing piece of evidence is the elucidation of the mechanism that mediates the choanoflagellate amoeboid switch, and whether it presents any similarity to the mechanisms that control animal cell differentiation. We now state this caveat explicitly in the conclusion:

“The nature of the confinement-sensitive pathway in choanoflagellates remains unclear. In the future, elucidation of this pathway will be necessary to test the hypothesis of homology between the choanoflagellate amoeboid switch and animal cell type differentiation mechanisms.”

Finally, the authors say that the switch between flagellated and crawling cells in choanoflagellates is triggered by particular size-related stress. However, it is difficult to imagine that animals evolved under such a type of stress.

The single-celled ancestors of animals, if they were benthic and occupied silts (as modern choanoflagellates do, alongside multiple amoebae and other protists (Leadbeater, 2015; Tikhonenkov and Mazei, 2006)) likely encountered compressive stress as part of their environment. As multicellularity evolved and became obligatory, we agree with the reviewer that such exogenous compressive stress became less likely as an inducer of cell type differentiation. However, cells within animal bodies frequently encounter confinement within the organism, due to the density of extracellular matrix or pressure from neighboring cells (Lomakin et al., 2020; Venturini et al., 2020). This type of “internally generated” compressive stress might have already existed at the incipient stages of the evolution of multicellularity. Indeed, a recent study from our lab has shown that cells within *S. rosetta* rosette colonies exert significant compressive forces onto each other (Larson et al., 2020). Thus, we posit that compressive stress continuously existed as a stimulus in the animal stem-lineage, although its source changed, during evolution, from the external environment to the organism itself (with presumably a phase of overlap when multicellularity was facultative). We have added a discussion of this point in the Discussion.

It may be interesting to discuss whether the authors have tried to see if alternative sources of stress induce transition to amoeboid states or, alternatively, discuss particular hypotheses about which kinds of stress might trigger this response.

We share the reviewer’s interrogation. Indeed, before we found that the amoeboid switch was induced by confinement, we originally thought it was caused by bacterial biofilms, and were surprised to discover – after multiple control experiments – that confinement was the actual (and sole) inducer. The behavior of *S. rosetta* has been studied by us and others in response to a large number of stimuli, including a broad range of bacterial species, different oxygen levels, different pH, nutrient status, light, and multiple other microbial eukaryotes. None of these conditions has been observed to induce the amoeboid switch so far. While we cannot rule out that other, confinement-independent stimuli might be able to induce the amoeboid switch (indeed, we would be excited if they were discovered), there is no indication of the existence of such stimuli in the currently available data. We now discuss this point:

“An open question is whether stimuli other than confinement can induce the amoeboid switch in choanoflagellates. […] While we cannot rule out the possibility that stimuli other than confinement might be able to induce amoeboid phenotypes (notably in species that have been studied less extensively than *S. rosetta*), it is currently unclear whether such other inducers exist.”

When discussing this, it may be worth considering an alternative: that choanoflagellates might be a side phylogenetic group having evolved specific characteristics and virtually lost amoeboid stages except for extreme situations like the ones shown here.

We acknowledge that amoeboid motility may be a minor part of the life history of modern choanoflagellates and might be somewhat reduced from an ancestrally more extensive version. We now address this point in the Discussion:

“The crawling behavior of the choanozoan ancestor might even have been more extensive than what we observed in modern choanoflagellates: indeed, most choanoflagellate species have secondarily lost some genes often involved in crawling motility, such as the integrin complex (Parra-Acero et al., 2020; Richter et al., 2018) and the transcription factors Brachyury (Sebé-Pedrós et al., 2016) and Runx (Brunet and King, 2017; Richter et al., 2018).”

However, two things suggest that amoeboid motility may be a far more important part of choanoflagellate life history than previously thought. First, confinement-induced amoeboid motility is found in nearly all choanoflagellates that we tested (Figure 6A-J, M), suggesting that it was present in their common ancestor and maintained by natural selection in all major modern choanoflagellate lineages. Second, we currently know very little about the environments in which choanoflagellates are most frequently found in nature. Planktonic environments are easiest to survey and have been the source of most choanoflagellate isolates, but choanoflagellates are frequently observed attached to surfaces (i.e. benthic environments) and are often detected by metagenomic surveys in environments, such as silts, that would expose them to confinement and presumably induce amoeboid motility. Text to addressing this point was included in the Introduction:

“Moreover, cell confinement is likely of ecological relevance for choanoflagellates, which have been detected in diverse granular microenvironments (soils (Geisen et al., 2015), marine sediments (McKenzie et al., 1997; Nitsche et al., 2007), sands (Tikhonenkov and Mazei, 2006), and silts (Tikhonenkov and Mazei, 2006)) whose pore sizes range from 1 mm to < 1 μm and extend below the range of typical choanoflagellate cell diameters (~2 to 10 μm) (Leadbeater, 2015).”

Indeed, our model choanoflagellate, *S. rosetta*, was isolated from a sediment core (where it may have been amoeboid), but has only been observed in its flagellate form in the laboratory until we realized that amoeboid motility could be induced under confinement.

We raise these points to make clear the fact that confinement may not be an extreme situation, but possibly rather a common event in the life histories of choanoflagellates.

The ancestor of metazoans would then had simply retained an ancestral (eukaryotic) capability to transition from flagellated to amoeboid states during the opisthokont life cycle without this capability being in any way related to volumetric stress but rather to particular environmental clues. Overall, all these ideas should be discussed in the context of eukaryotic evolution and toning down the potential implications.

We agree that this a possibility and we now explicitly discuss it:

“Alternatively, the alternation between amoeboid and flagellate forms might have been ancestrally transcriptionally regulated, possibly by other signals than confinement, and have evolved toward a “streamlined” post-transcriptional stress response in choanoflagellates. Study of additional holozoans will be crucial in distinguishing between these hypotheses.”

3) The authors use the standard definition of blebbing: actomyosin cortex breakage and healing concomitant with production of round protrusions devoid of actin.

Blebs are defined as F-actin-free cellular protrusions that form by delamination of the plasma membrane from the underlying actomyosin cortex (review in (Charras and Paluch, 2008; Fritz-Laylin et al., 2018; Paluch and Raz, 2013) for example). Blebs can be generated by two possible mechanisms: breakage and healing of the cortex, or detachment of the membrane from an intact cortex under the effect of contractile forces (reviewed in (Charras and Paluch, 2008; Paluch and Raz, 2013)). We are agnostic as to what mechanism dominates in choanoflagellates, which is why we did not discuss this point in the first version of the manuscript. Indeed, these mechanisms can be difficult to differentiate even in well-established model systems such as HeLa cells, which are 3 times larger (in diameter) than choanoflagellates and have proportionately larger blebs, as even in those large cells cortex breakage is often too transient or too small to be imaged (see the discussion of this point in the reviews by Charras and Paluch and Paluch and Raz). We now explicitly mention this limitation of our study:

“In animal cells, bleb formation has been proposed to be initiated by two possible mechanisms: breakage of the actomyosin cortex followed by detachment and blistering of the plasma membrane overlying the wound, or detachment of the plasma membrane from an initially intact actomyosin cortex (Charras and Paluch, 2008; Paluch and Raz, 2013). These two mechanisms have often proven difficult to experimentally distinguish in animal cells (reviewed in (Paluch and Raz, 2013)) despite their relatively large size, and it is currently unclear which of these two mechanisms underlies the initiation blebbing in choanoflagellates.”

Whether they are formed by one or the other of these two possible mechanisms, blebs are defined as F-actin-free protrusions that form by detachment of the plasma membrane from the actomyosin cortex (independently of whether this cortex is damaged or intact). We have added live imaging data to document the presence of blebs of that definition in confined choanoflagellates, and notably present a detailed time-lapse series with accompanying line scans in a new Figure 3.

The paper provides insufficient data to support the claim that choanoflagellate cells defined as "amoeboid" based on the lack of microvilli undergo this form of blebbing. Providing additional examples and/or quantitative analyses of the data would strengthen the paper. Specifically:a) Only a single cell with LifeAct is shown (Figure 2C, with what looks to be the same cell in Video 7). Additional examples would be welcome. However, this cell has high levels of septin overexpression: could this be interfering with the native phenotype?

Although a number of technological challenges prevented us from collecting additional images of LifeAct-expressing cells at the time of submission, these have been overcome during the past six months and we now providing images of multiple LifeAct-expressing cells that are not overexpressing any other fluorescent protein (such as septin2-mTFP).

For the reviewers’ interest, we summarize here the steps that enabled these experimental breakthroughs also reported in the Materials and methods:

To generate additional images and eliminate the potential confounding effect of septin2-mTFP overexpression, we set out to devise an alternative strategy, and to generate a stable LifeAct-expressing strain by co-transfecting LifeAct with a Pac puromycin resistance gene and selecting for resistant cells. First attempts were based on the design of a fusion protein Pac-P2A-LifeAct-mCherry, with a self-cleaving peptide P2A between Pac and LifeAct. While this strategy has successfully produced mTFP-expressing *S. rosetta* strains in the past, it failed with LifeAct, potentially due to imperfect P2A cleavage and/or abnormal folding of the fusion protein.

We then attempted instead to produce a construct with Pac and LifeAct-mCherry under the control of two distinct promoters. Initial tests with mCherry as a reporter allowed us to identify a suitable backbone for dual expression, in which we inserted LifeAct. Transfection of this construct allowed us to generate a LifeAct-mCherry-expressing strain, which has now been stable over more than five passages. This has allowed us to generate a large imaging dataset of LifeAct-mCherry-expressing cells under confinement, and we have now imaged blebbing in 15 distinct cells, of which we present four in Figure 2C-K, Figure 3, and Videos 5, 6 and 7. These new data confirm that the cellular protrusions of amoeboid cells are blebs, which are F-actin-free when they form but are reinvaded by F-actin before retraction. Moreover, this dataset eliminates the possible confounding effects of co-expression of septin2-mTFP.

Figure 3K looks very different from Figure 2Q.

We assume the reviewer refers here to the difference between fixed cells stained with phalloidin (Figure 2Q) and live cells expressing LifeAct-mCherry (Figure 2K). (Figure 3K shows escaping cells visualized by DIC microscopy and epifluorescence imaging of cytoplasmic mTFP, and was thus not meant to be comparable to Figure 2Q.)

To address this point, we have now improved our phalloidin/FM143FX staining protocol in a way that allows clearer visualization of cortical F-actin and is much more directly comparable to the distribution of LifeAct-mCherry. Specifically, we have found that the inclusion of acetone as a permeabilizer during fixation (inspired from the *S. rosetta* immunofluorescence protocol) considerably improved F-actin preservation and visualization of the cortex. We present this new dataset, combining micrographs of confined and unconfined cells with corresponding line scans of F-actin, in Figure 2L-S. We hope these new data now make the presence of cortical actin unambiguous and are much more directly comparable to the live imaging data in Figure 2C-K.

Is septin localizing to the membrane?

As described above, we have generated a new strain that expresses LifeAct-mCherry in the absence of septin2-mTFP overexpression and now focus our analyses on this new dataset. While the localization of septin is beyond the focus of this manuscript, the reviewers may be interested in the analyses in (Booth et al., 2018) in which septin2 was found to localize to the membrane and notably to a patch at the basal pole of cells.

The WT cells shown in P do not show cortical actin.

Thank you for pointing this out. As detailed above, our FM143FX/phalloidin co-staining protocol has since been improved, and we hope the new data in Figure 2L-S now make the presence of cortical actin unambiguous.

It seems likely worthwhile repeating this experiment with LifeAct but without septin overexpression. Additionally, linescans to quantitate the cortical actin levels before, during, and after cortical breakage would provide quantitative support, particularly with a membrane stain, or at the very least, the septin localization.

Thank you for this suggestion. We have now performed line scans to quantify levels of cortical actin during bleb formation, maturation and retraction. We present these results, alongside spatially and temporally detailed images of the corresponding bleb, in Figure 3.

To visualize the plasma membrane, we simultaneously imaged the same cell using DIC microscopy. (As an aside, limitations in our current protocols for choanoflagellate transfection prevent us from expressing more than two transgenes in a strain, and we were already expressing transgenes LifeAct and Pac, preventing us from adding in a fluorescent membrane marker. We have found that live dyes – such as FM dyes – bleach too quickly to allow extended imaging of blebbing.)

b) The phalloidin staining in Figure 2 and Figure 2—figure supplement 1 is not particularly convincing in terms of the presence of cortical actin. Why do the cells in Figure 2P and W look different? The additional examples also show weak actin staining.

As described above, we have now improved our phalloidin/FM464FX staining protocol, allowing us to better visualize cortical actin. Please see Figure 2L-S.

c) Figure 2S and V show myosin staining which does not appear to be localized to the cortex. More than a single cell should be shown, along with quantitation by line scan analysis to support the claim of cortical localization.

Thank you for pointing this out. To improve our analyses of myosin localization, we have generated a fluorescently-tagged myosin II (MRLC-mTFP) and used it for quantification of localization by line scan analysis (see Figure 4).

Upon confinement, MRLC-mTFP condenses into a set of foci and fibers, some of which are cortical and some of which are cytoplasmic. As a consequence of this condensation, the fraction of cortical myosin II increases. This is supported by three line of evidence: (1) timelapse imaging of two transfected cells before and during confinement, which shows condensation of myosin II in less than a minute (Figure 4B); (2) quantification of cortical myosin II staining in several batches of unconfined, 2 um-confined, and 1 um-confined cells imaged by confocal microscopy (Figure 4C-D); (3) two examples of cells that are half-confined under a micropillar. The confined part of the cell displays cortical myosin II fibers and spots, while the unconfined part shows diffuse cytoplasmic staining (Figure 4E). The myosin II spots and fibers seen in confined cells undergo complex intracellular movements, possibly due to contraction of the network (Figure 4-video 1, Videos 8 and 9).

Finally, we have also imaged MRLC-mTFP dynamic in blebbing cells, and found that, as for F-actin, expanding blebs are devoid of myosin II, but become encased in myosin II before retraction (Figure 4F, Figure 4-video 1 and Video 8).

d) Definition of amoeboid seems problematic as many of the images show "amoeboid" cells with what look to be microvilli: 2C-K, 2U-W, 3A-F

We agree – microvilli persist longer after flagellar retraction, and a variable amount of time can elapse before they are fully scattered and/or retracted. We have updated our definition of amoeboid cells to refer to cells which display blebbing and have retracted their flagellum, regardless of the presence or absence or microvilli. The Introduction now states:

“While microvilli initially persist in amoeboid cells, they undergo progressive scattering and, eventually, resorption over the following minutes (Video 2).”

e) The text describing the phalloidin staining is a bit circular as it assumes blebbing to interpret the staining patterns, but then uses the staining pattern as confirmation of blebbing.

Thank you for raising this issue. Our goal was to investigate the nature of cellular protrusions using two different approaches. New experiments and new data have provided independent validation that the cellular protrusions are blebs and, we feel, removed any circularity in the interpretation. Namely, our improved FM 1-43FX/phalloidin stainings now make clear that the protrusions we interpret as retracting blebs are actin-encased, rather than actin-filled (following the distinction emphasized above by the reviewers; see Figure 2P-S). This allows their unambiguous identification as retracting blebs rather than pseudopods and validates the findings from LifeAct-expressing live cells.

4) The authors indicate that the actin in amoeboid choanoflagellates undergo retrograde flow. The authors show a single cell imaged with widefield fluorescence (Figure 3F-G, Video 9). Typically retrograde flow is visualized using TIRF microscopy to show the movement of individual actin filaments within a network. The data presented makes it difficult to evaluate whether this is the retraction of the entire cortical network, or flow of actin filaments within a network. To support a claim of retrograde flow, additional data and analysis should be provided. Moreover, the concept of retrograde flow in the context of blebbing motility needs to be explained more fully in the text.

We agree with the reviewer that TIRF microscopy would be necessary to properly test for the existence of retrograde flow. Due to limited availability of shared microscopes during the Covid-19 pandemic, we could not perform TIRF microscopy on amoeboid choanoflagellates. We have thus removed the discussion of retrograde flow, and the corresponding data, from our manuscript. We hope to be able to address this issue in future studies.

5) The microtubule inhibition experiments involve treating cells for 36 hours with a non-standard microtubule inhibitor. Due to the possibility of off-target effects, the authors should repeat this experiment with a second microtubule inhibitor to cross-validate the result.

Thanks for raising these points. We tested both colchicine and nocodazole and found neither resulted in appreciable microtubule depolymerization after 24 hours. We have therefore decided to remove this dataset and corresponding discussion from the manuscript, and have only kept the tubulin immunostainings in flagellate and amoeboid cells (Figure 4—figure supplement 2). We hope to be able to identify fast and specific methods to depolymerize choanoflagellate microtubules in the future, and to be able to address this question in future studies.